

# Modelling the Wind Turbine Inflow with a Reduced Order Model based on SpinnerLidar Measurements

Anantha Padmanabhan Kidambi Sekar, Marijn Floris van Dooren, Andreas Rott, and Martin Kühn

ForWind, Institue of Physics, University of Oldenburg, Küpkersweg 70, 26129 Oldenburg, Germany

**Correspondence:** Anantha Padmanabhan Kidambi Sekar (anantha.kidambi@uni-oldenburg.de)

**Abstract.** Preview measurements of the inflow by turbine-mounted lidar systems can be used to optimise wind turbine performance by increasing power production and alleviating structural loads. Here, we apply Proper Orthogonal Decomposition (POD) to the line-of-sight wind speed measurements of a SpinnerLidar obtained from a large eddy simulation of a wind turbine operating in a turbulent atmospheric boundary layer. The aim of this work was to identify the dominant POD modes to derive
a reduced order representation of the turbine inflow without making strong assumptions about the flow field. This dimensional reduction is a first step towards the development of a reduced order inflow model (ROM) that offers a trade-off between wind field reconstruction techniques requiring flow assumptions and more complex physics-based representations. We found that only a few modes are required to capture the dynamics of the wind field parameters commonly used for lidar assisted wind turbine control such as the effective wind speed, vertical shear, directional misalignment. A possible interpretation of the modes
is presented by direct comparison with these wind field parameters. Evaluating six different metrics in the time and frequency domains related to the spatial, frequency domain and energy quantities, we find that a 10 mode ROM could accurately describe most spatio-temporal variations in the inflow. The reduced order modelling was accomplished using the inherent volume averaging property of lidar devices that attenuates high frequency turbulence with lower importance for the overall turbine response thus allowing significant data compression. Based on the models inflow wind field reconstruction performance, this method
has potential use for lidar-assisted control, loads validation and turbulence characterisation.

## 1 Introduction

With recent advances in the field of lidar technology for wind energy applications, scanning and measurement of the inflow of wind turbines have attracted greater attention. Turbine-integrated lidar systems can scan wind fields upstream of the turbine, allowing these measurements to be incorporated into turbine operation and control. Using this information as an input, turbine
performance can be improved in terms of power production and reducing the structural loads by feed-forward lidar-assisted control. Substantial amount of research has been done on lidar-assisted wind turbine control following advances in photonics-based communications for fibre-based lidar technologies that emerged during the early 2000's (Harris et al., 2007). Simley et al. (2018) provide an extensive overview of research on lidar inflow measurement based wind turbine control strategies. Turbine-mounted lidar measurements can also be used for loads validation (Dimitrov et al., 2019) and characterization of tur-
bulence upstream of the rotor (Peña et al., 2017).





The wind data quality extracted by the lidar depends upon the quality and beam scanning strategy of the lidar device itself. Modern lidar systems can perform very fast scanning measurements to capture the wind field with more detail. Such lidars represent an improvement over the multiple fixed-beam systems commonly found in commercial devices as they are instead outfitted with beam-steering mechanisms capable of moving and refocusing the laser beam to a predefined point or a scanning pattern in space (Mikkelsen et al., 2008). With such advanced devices available, it is possible to measure the inflow of a large wind turbine with high spatial and temporal resolutions. Due to the trend toward larger rotors, local wind field variations affect the turbine dynamics more strongly, hence it is necessary to measure wind fields over the entire rotor area.

Due to lidar's inherent line-of-sight limitations, wind field reconstruction (WFR) methods are required to extract even relatively simple parameters such as effective wind speed, direction and shear. Two types of WFR models can be found in the literature, i.e. static and dynamic WFR methods (Borraccino et al., 2017). In the static approaches (Borraccino et al., 2017; Kapp, 2017), wind fields are assumed to be stationary for a certain averaging period and spatial flow assumptions are made to determine relevant wind parameters. These models are adequate for power performance measurements as they well estimate the averaged wind characteristics. Kapp (2017) derived a five-parameter representation (i.e. the effective wind speed, and horizontal and vertical shear and directional parameters) from lidar measured wind fields by combining planar measurements at two different upstream distances. Taylor's Hypothesis (Taylor, 1938) is assumed between the measurement planes to resolve the ambiguity in shear and direction. Borraccino et al. (2017) introduced a static method based on a model-fitting technique and a least-squares solver to robustly reproduce ten-minute averaged wind parameters. This model was validated using full-scale measurements performed with two commercially available lidar devices ( the Avent five-beam demonstrator and the ZephIR dual-mode lidar). In the dynamic reconstruction methodologies, both spatial and temporal variations of the wind fields are taken into consideration. In Raach et al. (2014), a 3D model based dynamic WFR technique was presented by combining the static model presented in Schlipf et al. (2012) with Taylor's frozen turbulence hypothesis. Towers and Jones (2016) described a dynamic reconstruction methodology extracting two-dimensional horizontal wind fields at hub height from a pulsed lidar system with two fixed beams using an unscented Kalman filter with a lower order dynamic model to fill in the non-measured 2D velocity components of the wind fields. Guillemin et al. (2018) presented means of extracting real time 3D wind field parameters using a recursive weighted least-squares method validated with simulated pulsed lidar measurements. Kidambi Sekar et al. (2018) investigated the performance of LINCOM (a fast Navier-Stokes physics-based solver) to instantaneously reconstruct the local 3D velocity components from line-of-sight measurements of a SpinnerLidar on a spherical measurement plane upstream of the rotor.

Taylors hypothesis is assumed in the WFR methods proposed in Raach et al. (2014) and Kapp (2017): however, it does not hold in the turbine induction zone and complex inflow situations. Borraccino et al. (2017) used also a static method to provide ten-minute wind statistics rendering the method unusable for control and loads validation. The 2D reconstruction method suggested in Towers and Jones (2016) also assumes that measurements along all the beams are available simultaneously. Consequently, the performance of the models is limited for either reconstructing high frequency information (on the order of several seconds) or in situations where flow simplification assumptions are invalid, i.e. complex terrain or wind turbine wakes.

One alternative method is to accurately represent the inflow without making strong assumptions concerning the wind field. With recent advancements in scanning lidar technology such as the DTU SpinnerLidar (Mikkelsen et al., 2013; Herges et al.,



2017), which provides inflow scans with unprecedented spatio-temporal resolutions, it is possible to apply reduced order modelling techniques to real-time lidar inflow measurements. Using such high resolution data directly as a control signal is not feasible. A crucial step towards a lidar-based turbine control is a reduction of the measurement data to a few key variables, which still capture the most important spatio-temporal flow variations in the wind.

There are multiple ways to extract dynamic flow field information in fluid dynamics. One straightforward approach involves decomposing the flow field into a collection of spatial modes of which the most dominant are used to create a reduced order model (ROM) representation of the flow. Proper Orthogonal Decomposition (POD) is such an dynamic approach. This technique describes a velocity field as a linear combination of modes containing spatial information about the flow and time varying weighing functions defining the evolution of the flow field in time (Berkooz et al., 1993; Holmes et al., 2012). Mathematically,

the POD method calculates deterministic orthogonal basis functions for representing a spatio-temporal field. The decomposition is unbiased because it does not look for prior information and the basis functions are obtained from the dataset itself, in contrast to other techniques. As the modes themselves are orthogonal, this method is suitable for constructing reduced order models by directly truncating higher modes or using Galerkin projection to capture dominant flow physics and to implement model based closed-loop flow control (Taira et al., 2017). The dominant structures obtained via POD decomposition are repre-

sentative of the coherent motions in the wind flow (Holmes et al., 2012).

For wind energy applications, POD have been used to develop and understand dynamic wake models (Bastine et al., 2015; Andersen et al., 2017). Saranyasoontorn and Manuel (2005) applied POD to study inflow wind fields based on stochastic wind field simulations. They concluded that the inflow of a wind turbine can be represented by a very small number of POD modes for the longitudinal wind component. This was confirmed by Kidambi Sekar and Kühn (2017) where the methodology was

applied to full field SpinnerLidar measurements. A POD based reduced order model offers a trade-off between the simplicity of traditional WFR methods requiring flow assumptions, and more complex solvers using the line-of-sight measurements as a boundary condition for solving the Navier-Stokes equations.

The objective of this paper is to introduce a POD based dynamic WFR methodology that does not require strong assumptions about the reconstructed wind field. We aim to identify the dominant inflow spatial modes that can be used to obtain a lower or-

der model representation of the velocity field that is scanned with a spinner based lidar while retaining most of its information. To this end, we apply the POD method to virtual SpinnerLidar inflow measurements of a wind field taken from a large eddy simulation (LES) to have a realistic representation of the atmospheric boundary layer flow. The POD modes of the inflow field are estimated and used to reconstruct a reduced order model of the original flow using kinetic energy as the filter. Our second aim is to assess the quality of the WFR method based on relevant information concerning the turbine inflow. The evaluated

metrics include the effective wind speed, vertical shear and horizontal misalignment and turbulence spectra in the fixed and rotating reference frames. Additionally, we aim to interpret the modes based on their relationship to the defined metrics. The efficiency of the resulting ROM in capturing the dominant dynamics of the inflow while allowing for data compression is evaluated.

The article is structured as follows. The measurement device and LES simulations are described in Subsection 2.1 and Subsec-

tion 2.2. Subsection 2.3 describes the POD theory. The application of the POD method and the results are described in Section



3 where the method is applied to the LES simulations. POD based reconstruction of the original velocity fields is introduced followed by a quantitative analysis of the reduced order model based on wind field metrics. The performance of these metrics are investigated and compared against a reference. Section 4 discusses the results and Section 5 presents the conclusions.

## 2 Methods

To obtain a realistic wind field dataset to investigate the inflow to a wind turbine and create a reduced order model representation we employ virtual SpinnerLidar data derived from high fidelity LES. The SpinnerLidar specifications and working principles are presented in Subsection 2.1. The LES providing the wind field data are explained in Subsection 2.2 along with the Lidar Simulator (LiXim). In Subsection 2.3, the ROM is introduced based on the POD methodology while Subsection 2.4 describes the method used to extract and compare wind field parameters.

### 2.1   SpinnerLidar

The SpinnerLidar (Mikkelsen et al., 2013; Herges et al., 2017) is a modified ZephIR 300 continuous-wave Doppler lidar with a 2D scan head developed by the Technical University of Denmark (Fig. 1). This lidar can perform performing 2D measurements on a spherical surface of the radial line-of-sight wind speeds in front of a wind turbine with very high spatial and temporal resolutions. The device consists of two rotating prisms deviating the lidar's focused beam by an angle of $15°$ while rotating

at a fixed ratio of 7 to 13. The resulting scan pattern movement creates a fast rosette trajectory covering a large area with a quasi-homogeneous spatial resolution. The lidar is capable of providing 2D wind field scans at a temporal sampling rate of 1 Hz with a variable focal distance from 10 m to 150 m (albeit with a constant opening angle of $30°$). It can be set to sample a maximum of 500 radial line-of-sight measurements points distributed over each completed scan trajectory (Fig. 2).

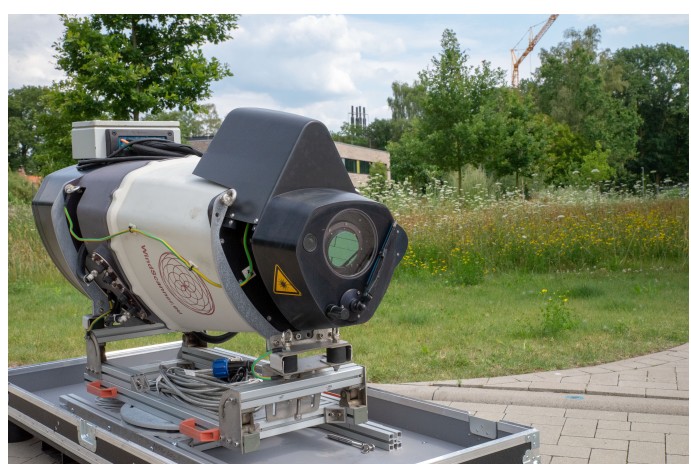

**Figure 1.** The SpinnerLidar with its mounting platform at ForWind - University of Oldenburg.





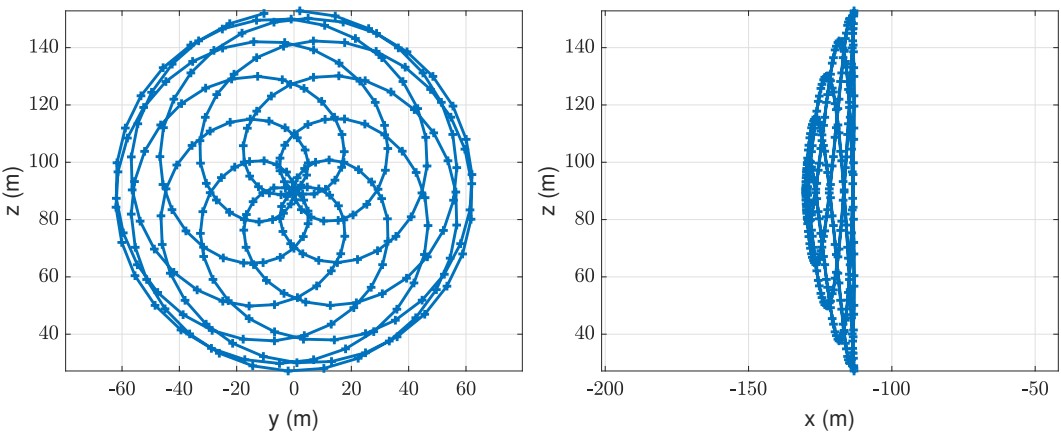

**Figure 2.** The rosette scanning pattern of the SpinnerLidar shown from two perpendicular views.

We note that the lidar can only provide wind speed measurements along the beam direction (so called the line-of-sight ($v_\mathrm{los}$) wind speeds): usually referred to as the 'cyclops dilemma' in literature (Schlipf et al., 2011). The line-of-sight $v_\mathrm{los}$ speed is expressed as a projection of the three wind speed components on the line-of-sight, as described by Harris et al. (2007) and shown in Eq. (1):

$$v_\mathrm{los} = \begin{pmatrix} \cos(\alpha)\cos(\delta) \\ \sin(\alpha)\cos(\delta) \\ \sin(\delta) \end{pmatrix} \begin{pmatrix} u \\ v \\ w \end{pmatrix}. \tag{1}$$

Here, $\alpha$ is the azimuth and $\delta$ is the elevation angle of the horizontal and vertical direction of the focused laser beam, respectively, both varying from -30° to 30°. The quantities $u$, $v$ and $w$ are the longitudinal, lateral and vertical wind components, respectively. To calculate the projected longitudinal component of the wind speed ($u$) from the inversion of Eq. (1), the lateral velocities ($v$ and $w$) are assumed to be zero as the longitudinal component usually dominates ($u \gg v, w$) for small yaw misalignment and tilt angles.

Another important property of continuous-wave lidar measurements is the probe length averaging effect. The line-of-sight measurements of the lidar are not collected at a single point in space, but are actually a weighted average over a thin cylindrical volume along the beam direction approximated as a Lorenzian function (Sjöholm et al., 2009), as the laser beam cannot be perfectly focused at an infinitesimal point in space. Most of the laser signal is reflected at the focal distance, but contributions from the vicinity of the focal point also exist. This results in an attenuation of turbulence. Equation 2 obtains $v_\mathrm{los}$ as a weighted
average of the scalar product of the local wind speed vector over the distance $s$ relative to the focus length $f$ along the line-of-sight direction and the unit vector $\mathbf{n_{los}}$:

$$v_{los} = \int_{-\infty}^{\infty} \frac{1}{\pi}\left(\frac{\Gamma}{s^2 + \Gamma^2}\right)\mathbf{n_{LOS}} \cdot \mathbf{u}\big((s+f)\mathbf{n_{LOS}}\big)\mathrm{d}s. \tag{2}$$



The probe length is considered to be twice the half width at half maximum ($\Gamma$) which is the distance from the focal point at which the backscatter spectrum is reduced to half its peak power and depends quadratically on the focal distance (Fig. 3). The probe length depends on the laser wavelength $\lambda$ and the minimum beam width $a$ which are fixed parameters. The probe lengths are 0.13 m and 28.3 m at $f$ = 10 m, 150 m corresponding to the minimum and maximum achievable focal distance of the SpinnerLidar.

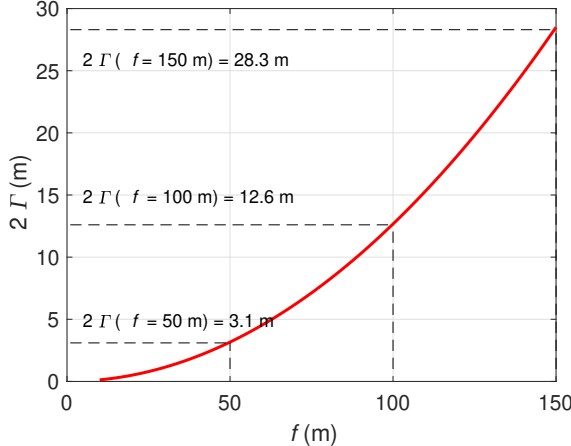

**Figure 3.** Full Width at Half Maximum ($2\Gamma$) function of the SpinnerLidar over its entire focal distance ($f$) range.

## 2.2 Large Eddy Simulations (LES) and the Lidar Simulator (LiXim)

The wind field information for the analysis was generated using LES. LES models are capable of accurately resolving the turbulent kinetic energy in the atmosphere making them a suitable candidate for simulating realistic inflow conditions while also providing a full 3D reference wind field for comparison and quality assessment. The LES data were obtained from simulations with the Parallelised Large Eddy Simulation Model (PALM). The PALM code is widely used for atmospheric boundary layer studies and works by solving the filtered, incompressible, non-hydrostatic Navier-Stokes equations. Further details of the model are available in Raasch and Schroter (2001) and Maronga et al. (2015). PALM employs the Schumann volume averaging approach and uses central differences to discretise the non-hydrostatic and incompressible Boussinesq approximation of the 3-D Navier–Stokes equations on a structured Cartesian grid. To model the effects of sub-grid scale turbulence on the resolved scale turbulence, a $1.5^{th}$ order closure is employed. Wind turbine and wind farm simulations using PALM have been performed, e.g., by Dörenkämper et al. (2015), Vollmer et al. (2016) and Bromm et al. (2018). The aeroelastic simulations are performed using the Fatigue, Aerodynamics, Structures and Turbulence code (FAST) v8, developed by the National Renewable Energy Laboratory (NREL). The code simulates the wind turbine as a combination of rigid and flexible bodies (Jonkman et al., 2005) and the aerodynamic forces are calculated via the AeroDyn module. Both PALM and FAST run simultaneously in an explicit loose two-way coupling. The velocities from the LES field are transferred to FAST, which subsequently calculates the lift and drag on the blade segments based on look-up tables of the airfoil characteristics. Next, the relative velocities of the





blades and their new positions are determined. This information is transferred back to PALM, where the forces are distributed
back into the flow field, where the induction zone and the wake are generated, as visualised in Fig. 4.

An unsteady, turbulent atmospheric boundary layer (ABL) with a single turbine was simulated using this framework. The wind
turbine model is the actuator line implementation of the NREL 5 MW reference turbine (Jonkman et al., 2009) with a rotor
diameter of 126 m and a hub height of 90 m with 63 blade sections. The simulation presented in this paper was performed using
the PALM code in its default settings. The full domain size for the simulations was 8188 m by 4092 m by 2048 m with a grid
spacing of 4 m and a sampling rate of 5 Hz. The large domain size ensures that the inflow of the turbine is unaffected by the
boundaries. To obtain unstable stratification, we performed a pre-run of 12 hours to allow the boundary layer to develop and
reach stationary flow. The instantaneous fields of the precursor simulation are mapped onto the main simulation via turbulence
recycling (Lund et al., 1998), which is applied at a distance of 3000 m from the inlet in the $yz$-plane which is large enough to
preserve all turbulent scales. The turbine is placed another 2000 m downstream at the coordinates (5000,1000,0). An unstable
ABL was simulated with a roughness length $z_o = 0.0175$ m and a friction velocity $u_* = 0.52$ m/s. The mean inflow wind speed
at hub height is 10.1 m/s with a turbulence intensity of 11.9% for a total duration of 3700 s.

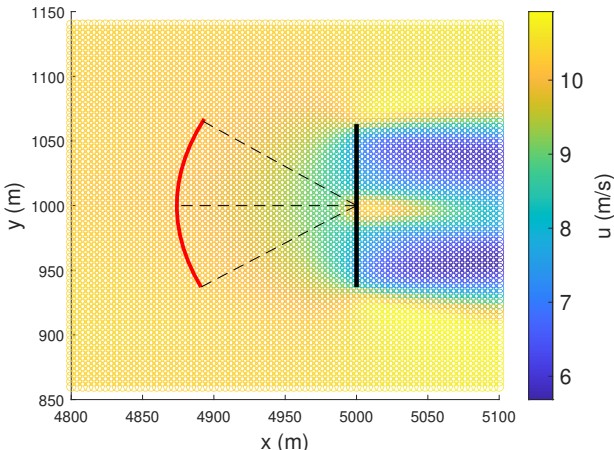

**Figure 4.** The averaged $u$-component at hub height extracted from the LES wind field for the total simulation duration. The black vertical
line indicates the position of the turbine rotor and the red line illustrates the SpinnerLidar measurements at $f = 126$ m. The low velocity
region behind the turbine is its wake.

To extract the SpinnerLidar measurements from the LES wind field, we used the integrated lidar simulation toolbox LiXim
(Lidar Scanner Simulator) developed by Trabucchi (2020). The SpinnerLidar is simulated in the LES wind field using the
LiXim simulator measuring at a focal distance of 126 m with a temporal resolution of 1 Hz for a total duration of 3700 s. The
lidar simulator emulates the lidar measurements inside the LES wind field while taking account of the volume averaging effect
and producing 312 measurement points distributed along the rosette scan pattern. The simulated lidar device is assumed to
have a horizontal orientation and the rotor tilt is not taken into account. The measurements are then interpolated onto a uniform





grid with a 3 m spacing resulting in 1261 grid points. The effective probe length in terms of the FWHM of the SpinnerLidar
when focused at a measurement range of 126 m ($1D$) is approximately 19 m as shown in Fig. 3.

### 2.3 Reduced Order Model based on Proper Orthogonal Decomposition

POD is a powerful numerical technique applicable to identify turbulent coherent structures from flow fields. The decomposi-
tion provides a set of time-independent orthogonal spatial modes with their respective time dependant coefficients. The basic
decomposition procedure is to sample the data, calculate the auto-covariance matrix, and solve the corresponding eigenvalue
problem which can then be used to construct an orthogonal basis. Several references describing the general POD methodology
can be found in Berkooz et al. (1993), Sirovich (1987) and Holmes et al. (2012). The practical implementation of this method
for application to SpinnerLidar measurements is discussed here. The line-of-sight measurements of the SpinnerLidar $v_{\mathrm{los}}(X,t)$
where $X = (x, y, z)$ are organised in a snapshot matrix $V$ defined as

$$V(X,t) = [V(X,t^1)V(X,t^2)\cdots V(X,t^n)] = \begin{pmatrix} V_1^1 & V_1^2 & \cdots & V_1^n \\ V_2^1 & V_2^2 & \cdots & V_2^n \\ \vdots & \vdots & \vdots & \vdots \\ V_N^1 & V_N^2 & \cdots & V_N^n \end{pmatrix}. \tag{3}$$

Where $n$ is the number of snapshots and $N$ is the total number of grid points in each snapshot. For a line-of-sight velocity field
$V(X,t)$, $||V(X,t)||^2$ is related to its turbulent kinetic energy. It is common to subtract the mean value to obtain the fluctuating
component $V'(X,t)$. POD decomposes the wind field $V'(X,t) = V(X,t) - \langle V(X,t) \rangle$ into a linear superposition:

$$V'(X,t) = \sum_{j=1}^{N} Z_j(t)\phi_j(X), \tag{4}$$

where $\phi_j(X)$ are called the spatial POD modes optimal with respect to the flow turbulent kinetic energy and $Z_j(t)$ are the time
evolving POD weighing coefficients. This solution is obtained by solving the eigenvalue problem of the covariance matrix
$R = V'(X,t)V'(X,t)^T$,

$$R\phi_j = \lambda_j\phi_j, \tag{5}$$

resulting in a set of eigenvectors $\phi_j$ denoted as POD modes and a set of corresponding eigenvalues $\lambda_j$ which can be ordered as
$\lambda_1 \geq \lambda_2 \geq \lambda_3...$ . The flow field can now be denoted as a linear combination of $N$ uncorrelated spatial modes:

$$V(X,t) = \langle V(X,t) \rangle + V'(X,t) = \langle V(X,t) \rangle + \sum_{j=1}^{N} Z_j(t)\phi_j(X), \tag{6}$$

where the $j^{\mathrm{th}}$ weighing coefficient is obtained as

$$Z_j(t) = \phi^T(X)V'(X,t). \tag{7}$$



$Z_j(t)$ contains the temporal gains determining the development of the POD modes in time. A reduced order representation of the flow field $\hat{V}(X,t) = \langle V(X,t) \rangle + \hat{V}'(X,t)$ can be obtained by sorting and assembling the modes by decreasing magnitude of the eigenvalues and by truncating the higher modes and retaining only $M < N$ modes where

$$\hat{V}'(X,t) = \sum_{j=1}^{M} Z_j(t)\phi_j(X). \tag{8}$$

For such a reduced order approximation, the POD modes of Eq. 5 are optimal with respect to the turbulent kinetic energy in the flow. Hence, they are a set of optimal modes with the least mean square error given by

$$\langle ||V'(X,t) - \hat{V}'(X,t)||_2^2 \rangle. \tag{9}$$

The spatial modes $\phi_j(X)$ contain information about coherent structures as the POD method can be seen as an energy filter that unravels the large spatial turbulent structures. However, it must be noted that these structures might not be actual physical structures present in the flow field, but merely a result of the mathematical operation. An important property of this method is that the POD modes are orthonormal and their temporal gains are uncorrelated (Holmes et al., 2012). With large sizes of the flow matrix being common in fluid flow problems, the dimension of the covariance matrix $R$ becomes quite high, thereby making the application of classical POD very time consuming. To avoid this, the method of snapshots proposed in Sirovich (1987) is used whereby the temporal covariance matrix $R = V'^{T}V'$ is solved to obtain the same dominant spatial POD modes. Due to the reduced computational and memory resources needed, the snapshot method is the most commonly used method for obtaining POD modes from flow data.

We can now apply the snapshot POD methodology to the line-of-sight velocity field $v_{\mathrm{los}}(X,t)$ obtained via SpinnerLidar measurements, truncate the higher modes and create a reduced order model of the flow. The order $M$ of the reduced model in Eq. (8) is crucial. Improper selection of $M$ might lead to a dimensional reduction that is either very large or very small and important flow field data may be lost. We aim to identify dominant inflow POD modes that will preserve the inflow's dynamic characteristics while providing a significant dimensional reduction.

## 2.4 Quantifying the accuracy of the Reduced Order Model

The main objective of any inflow model is to capture the inflow, along with its variations that significantly impact the wind turbine. Thus, the quality of ROM reconstruction should be evaluated with respect to wind field parameters that directly affect the turbine itself. Typical parameters required to implement a standard individual pitch controller are the rotor effective longitudinal wind speed $u_{\mathrm{eff}}$, horizontal misalignment $\delta_{\mathrm{h}}$ and vertical shear $s_{\mathrm{v}}$. To extract the wind field parameters from the SpinnerLidar line-of-sight measurements, we use a methodology based on Kapp (2017). As two measurement planes are needed to resolve the ambiguity between the shear and direction parameters, a simplified 3-parameter model consisting of the rotor effective wind speed $u_{\mathrm{eff}}$, horizontal misalignment $\delta_{\mathrm{h}}$ and vertical shear $s_{\mathrm{v}}$ is obtained by omitting the horizontal shear $s_{\mathrm{h}}$ and the upflow $\delta_{\mathrm{v}}$. The effective wind speed quantity $u_{\mathrm{eff}}$ is related directly to the turbine's dynamic response and power output. This is the main variable for selecting the operational condition of the turbine and an input for collective pitch control.





The vertical shear $s_\mathrm{v}$ is important for individual pitch control algorithms, whereby the controller compensates for vertical
shear present in the ABL by pitching the blades individually. The horizontal misalignment $\delta_\mathrm{h}$ is an important parameter for
determining the turbine's yaw setting. To quantify the reconstruction error associated with these wind field parameters, we
write:

$$\hat{Y}[v_\mathrm{los}(X,t)] = Y(t), \tag{10}$$

where $Y(t)$ is the set of rotor-averaged wind field parameters $u_\mathrm{eff}, \delta_\mathrm{h}, s_\mathrm{v}$ calculated by applying a 3-parameter function $\hat{Y}$ to
the line-of-sight velocities. Similarly, the wind field parameters for the reduced order model with $M$ modes can be represented
as:

$$Y^M(t) = \hat{Y}[v_\mathrm{los}^M(X,t)]. \tag{11}$$

The quality of the reconstructed wind field $v_\mathrm{los}^M(X,t)$ and the corresponding wind field parameter $Y^M(t)$ with $M$ modes can
be calculated by comparison to the original wind field parameters $Y(t)$. To evaluate the reconstruction efficiency, two different
error parameters are introduced as defined by Bastine et al. (2015). The error associated with the reconstruction of a wind field
parameter $Y(t)$ is defined as:

$$\epsilon_\mathrm{std}(M) = \frac{||Y^M - Y||_2}{||Y||_2} \qquad\qquad \epsilon_\mathrm{dyn}(M) = \frac{||(Y^M - \langle Y^M \rangle_t) - (Y - \langle Y \rangle_t)||_2}{||Y - \langle Y \rangle_t||_2}, \tag{12}$$

where $\epsilon_\mathrm{std}$ and $\epsilon_\mathrm{dyn}$ represent the standard and dynamic error respectively. These values respectively quantify the mean error
and the error associated with the fluctuations $Y(t) - \langle Y(t) \rangle$ in the wind field. The definition of the dynamic error is motivated
by the fluctuation of wind field parameters having has the largest effect on the loading and the fatigue on the turbine in contrast
to the mean wind field parameters. To investigate the relationship between the POD temporal gains $Z_j(t)$ and the wind field
parameter time series, the Pearson correlation coefficient $\rho$ is used:

$$\rho = \frac{\mathrm{cov}(Z_j(t), Y(t))}{\sigma_{Z_j(t)} \sigma_{Y(t)}}, \tag{13}$$

Here, $cov$ is the covariance of $Z_j(t)$ and $Y(t)$ and $\sigma$ are the respective standard deviations.

## 3 Results

The performance of the POD based reduced order model is tested for the virtual SpinnerLidar in the LES wind field. The
simulations are used as a benchmark for comparing and quantifying the accuracy of the inflow model based on different
reconstruction metrics such as the convergence of POD modes, eigenvalue distributions, velocity field reconstruction, wind
field parameterisation and turbulent spectra in the fixed and rotational frames of reference.



### 3.1 Convergence of the Proper Orthogonal Decomposition

The first test of the performance of the wind field reconstruction model is to check for convergence of the POD modes. Typically, the dominant high energy POD modes associated with the large-scale turbulent structures are well converged. To test the convergence of POD modes, velocity data over various numbers of timesteps are taken and the change of POD modes with time is examined. Ideally, the process is said to converge if the eigenvalues and POD modes shapes remain constant as the numerical solution progresses. Figure 5 illustrates the eigenvalues associated with the first six POD modes and their corresponding rates of change with respect to the number of samples. The magnitude of the first six eigenvalues vary up to a dataset size of $n = 1000$ samples and start to converge around $n = 3500$ samples due to the temporal correlations in the wind field. The eigenvalues rate of change of the falls below 5% for the first 6 modes calculated with more than 3000 samples. Figure 6 illustrates the change in the shape estimate of the third POD mode as a function of the number of samples used in the calculation. The basic structure of the mode is already visible with a sample size of 1700. The addition of more samples does not produce any significant changes in the shape of the modes. As the dominant modes obtained for the dataset are well converged, the use of POD to extract large scale structures is considered to be justified (Hekmati et al., 2011). Further on, we apply the POD method to the total sample duration of 3700 s.

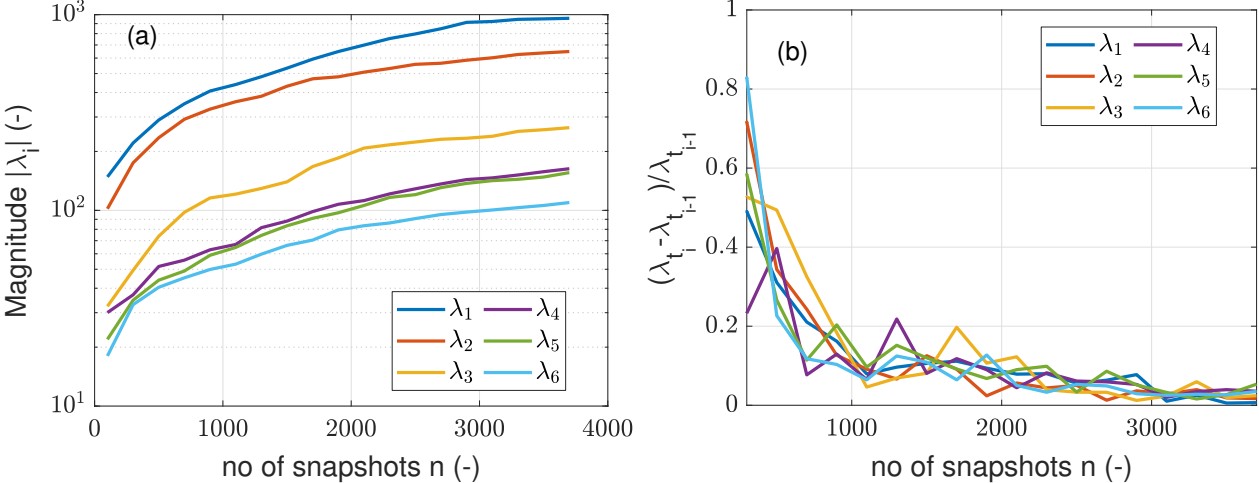

**Figure 5.** (a) Magnitudes of the first six eigenvalues and (b) rate of change of eigenvalues of the first six modes with respect to the number of snapshots for the same dataset.

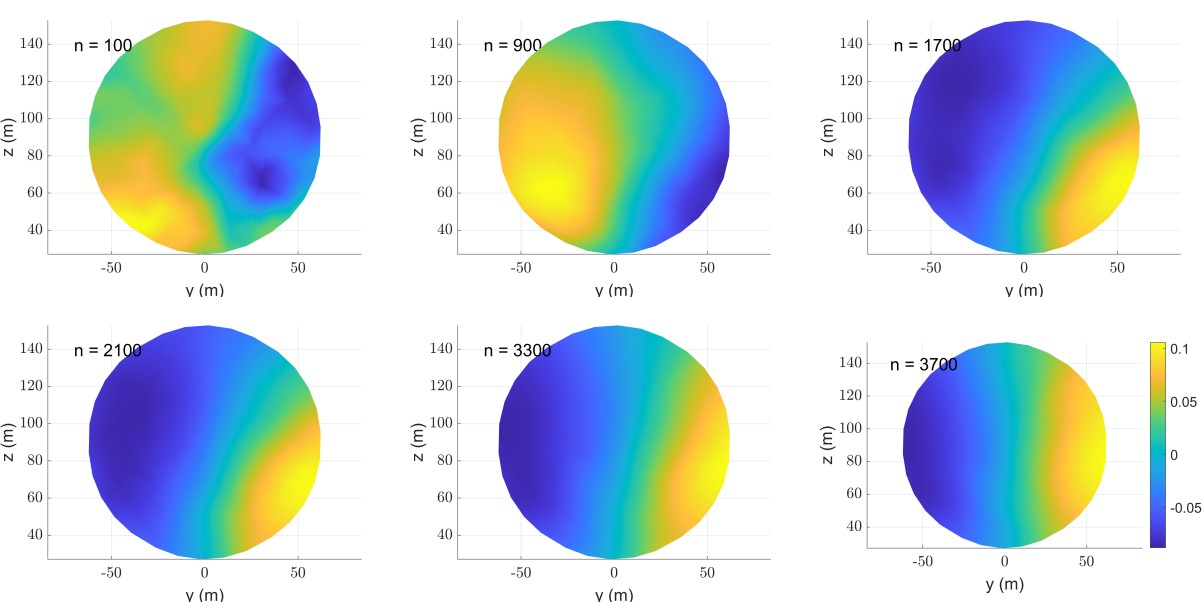

**Figure 6.** Shape of the third POD mode with ensemble sample size $n = 100$, $n = 900$, $n = 1700$, $n = 2100$, $n = 3100$ and $n = 3700$.

## 3.2 Application of the POD methodology to SpinnerLidar measurements

270 The eigenvalues and eigenvectors associated with the covariance matrix for the line-of-sight velocity field are calculated based on Eq. (5).

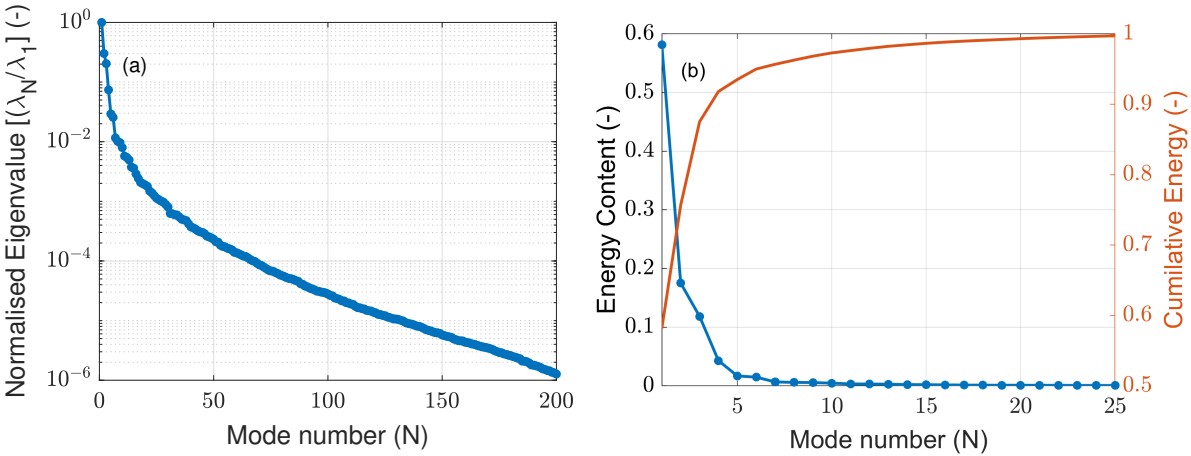

**Figure 7.** Eigenvalues and the energy contribution of each mode.

The normalised magnitudes of the eigenvalues $\lambda_j$ and the fraction of energy associated with each mode of the $v_{\mathrm{los}}$ measurements are shown in Fig. 7. The first POD mode contains 57.7% of the total measured energy while the second, third, and fourth

mode contribute 17.3%, 11.8%, and 4.3%. It is clear from Fig. 7, the first 10 modes contribute 96.6% of the measured turbulent

275 kinetic energy (TKE) while the first 100 modes accounts for 99.95%. Note that the total TKE measured by the lidar is not

equivalent to that in the wind field due to volume averaging induced turbulence attenuation of the line-of-sight measurement.

This may be surprising since in a turbulent flow, energy is distributed over the different scales and its representation might usu-

ally require an enormous number of POD modes. However, a lidar system acts as a low-pass filter for small scale turbulence

due to its volume averaging property. Hence, the small scale turbulence is filtered out and an accurate representation of the

280 remaining SpinnerLidar measured total TKE can be recovered with very few modes.

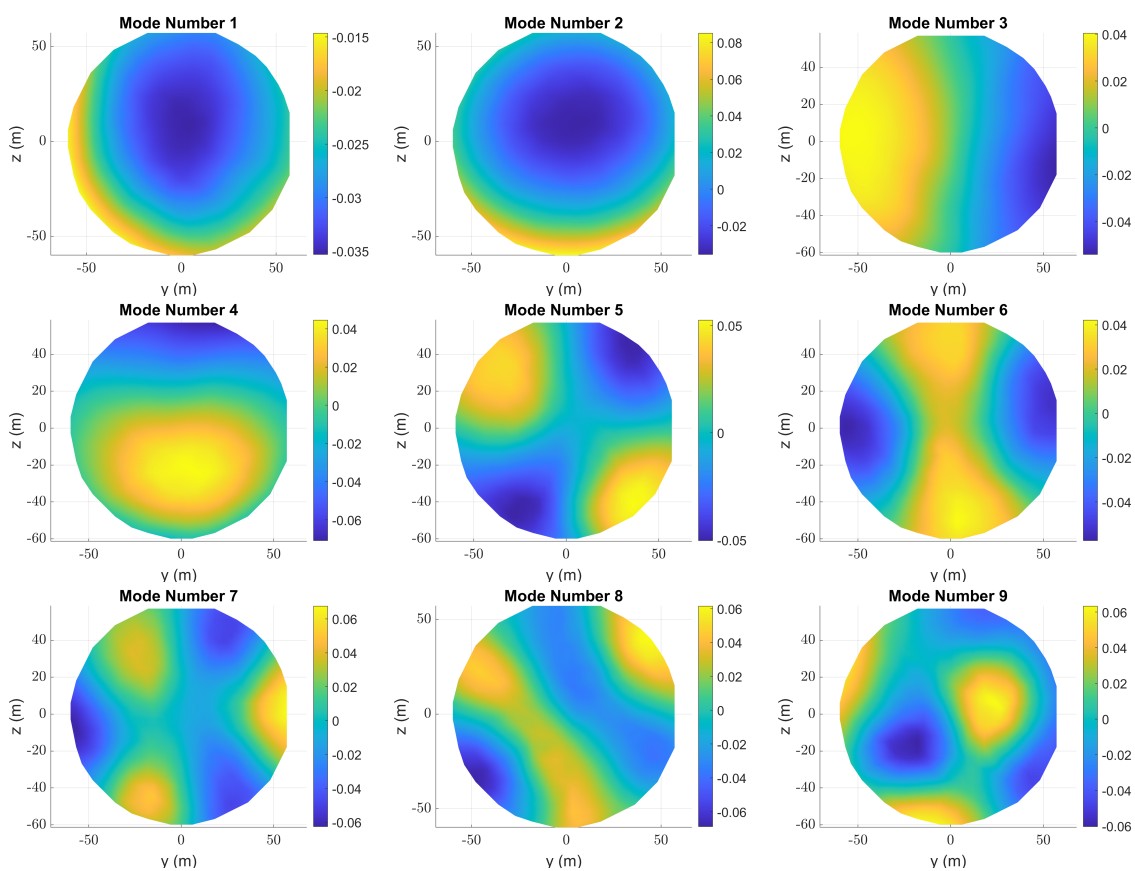

**Figure 8.** POD modes 1 to 9 of the line-of-sight ($v_{\mathrm{los}}$) measurements.

The first nine POD modes $\phi_j(X)$ are illustrated in Fig. 8. Well defined modes are obtained indicating convergence. The

modes exhibit clear structures trend towards smaller scales with increasing mode number as the modes are sorted based on the

energy content and the kinetic energy decreases with scale. Modes 1 and 2 have a similar mode shape with different gradients,

285 resulting from the variation in the line-of-sight velocities decreasing toward the edges of the scan pattern. The first two modes

do not exhibit clear symmetry around the rotor axis due to wind shear present in the ABL. Modes 3, 4 display variations in





the horizontal and vertical directions respectively. Modes 5 and 6 have quadrupole distributions while modes 7 and 8 show weak hexapole distributions. These observations exhibit weak statistical isotropy commonly observed in turbulent flows. As expected, the higher mode patterns become increasingly complex compared with the first few modes. It should be noted that the shape of the POD modes will differ based on the inflow conditions due to variable behaviour of the ABL under different stratification conditions.

### 3.3 Reconstruction of the velocity field

The extracted POD modes are used to reconstruct the velocity field based on Eq. (8) by choosing increasing values of $M$ (the number of modes used for reconstruction). A snapshot of the velocity field at an arbitrary time $t = 256$ s is reconstructed and illustrated in Fig. 9 for different number of modes along with its corresponding planar velocity reconstruction error in comparison to full lidar measurements. As expected, a more detailed wind field reconstruction is achieved with an increasing number of modes. For the reconstruction with $M = 1$ (first mode alone), only the mean wind speed of the measurements is obtained, as indicated by a constant velocity distribution over the whole measurement plane which is also supported by the $M = 1$ error plot with respect to the full lidar measurements. The addition of more modes into Eq. (8) adds more localised wind field information as the smaller wind field fluctuations which are contributed by the higher modes are taken into account. The velocity field reconstruction with ten modes shows close agreement with the untruncated full lidar measurements with reconstruction error effectively reducing to zero throughout the measurement plane. The high recovery of kinetic energy in the first few modes as discussed in Subsection 3.2 indicates that only a few modes are required to create a meaningful reduced order model capturing all spatial variations in the wind field.



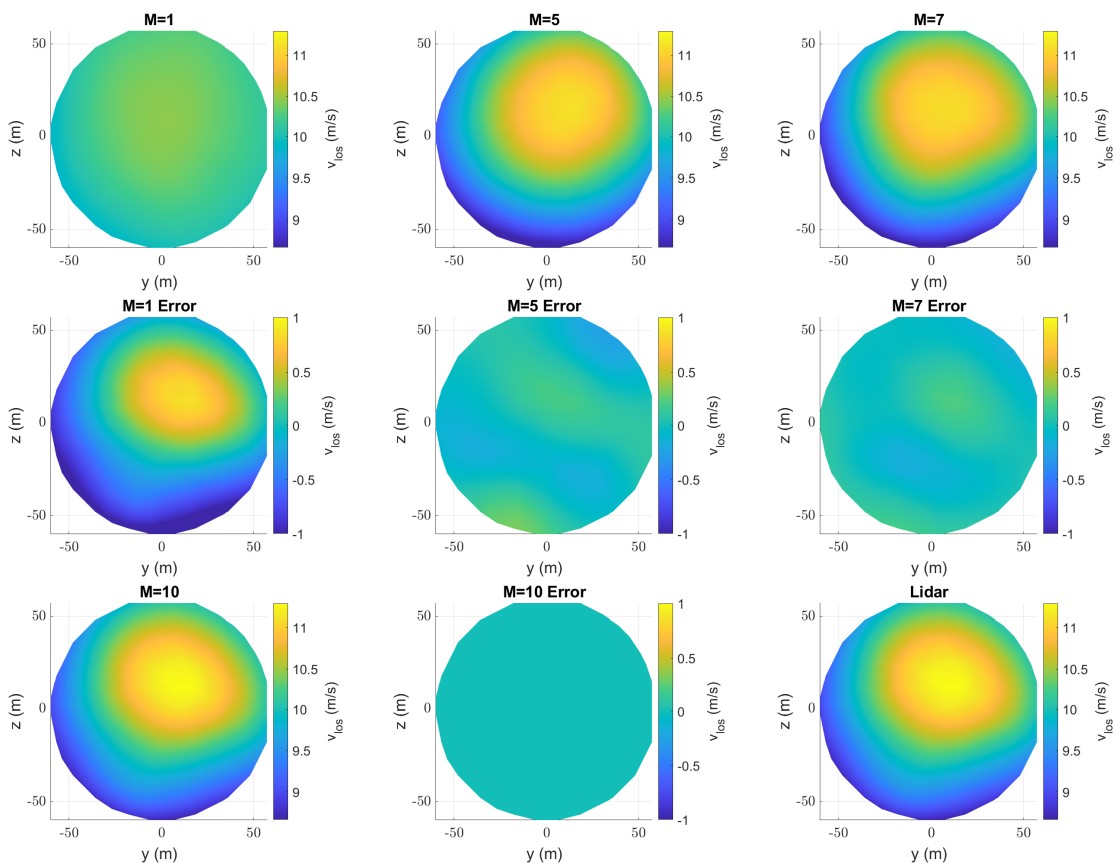

**Figure 9.** Reconstruction of the line-of-sight ($v_\mathrm{los}$) velocity snapshot at an arbitrary time instant $t = 256$ s with different number of modes ($M = 1, 5, 7, 10$) and their corresponding errors with respect to full lidar measurements (bottom right-hand plot captioned as lidar).

### 3.3.1 Time series reconstruction of the three-parameter wind field model

To quantify the reconstruction accuracy, inflow wind field parameters are extracted from the line-of-sight velocity distribution defined in Subsection 2.4. The results of the extracted wind field parameters from different ROMs are compared with each other and with the full lidar measurements. The three-parameter wind field model is applied to the SpinnerLidar simulations which will henceforth be referred to as direct determination. Various ROMs are created by truncation with $M = 1, 2, 3, 5$ and 10 modes and the associated wind field parameters ($u_\mathrm{eff}, \delta_\mathrm{h}, s_\mathrm{v}$) are calculated and compared with direct determination.



eawe
european academy of wind energy

WIND
ENERGY
SCIENCE
DISCUSSIONS

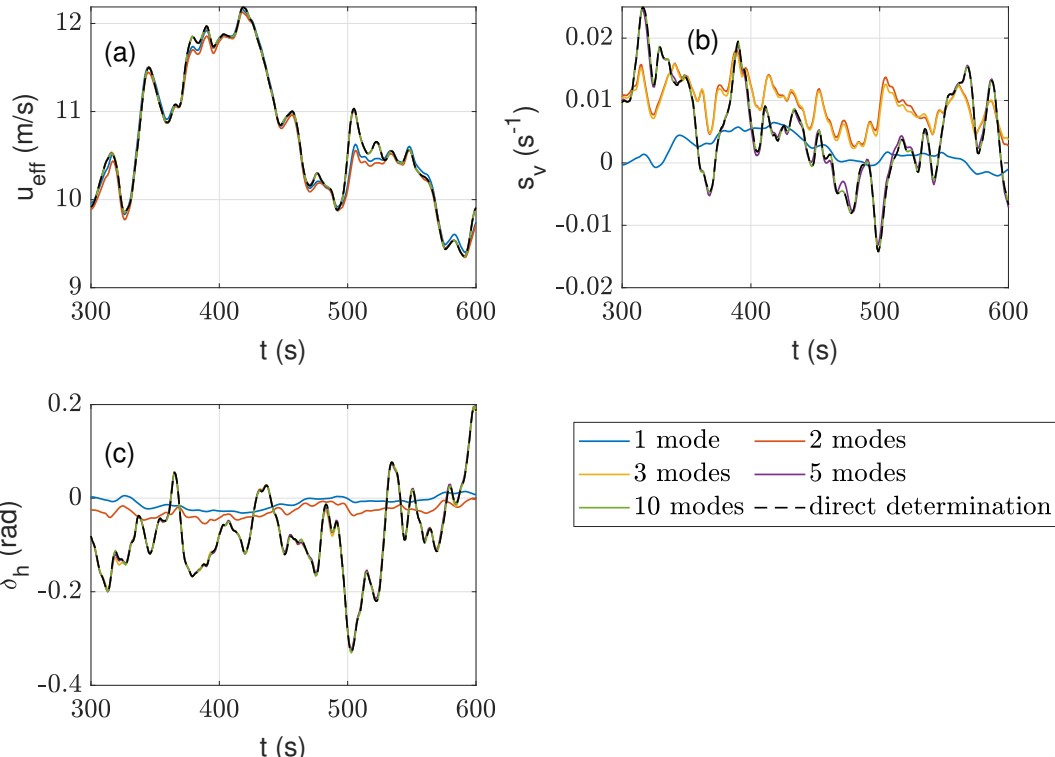

**Figure 10.** The time series of the wind field parameters $u_{\text{eff}}, s_{\text{v}}$ and $\delta_{\text{h}}$ extracted at a sampling rate of 1 Hz for a duration of 300 seconds for different POD reconstructions.

Figure 10 contains part of the time series of the three wind field parameters for different ROMs with $M = 1, 2, 3, 5$ and 10. For the reconstruction of rotor effective wind speed $u_{\text{eff}}$, the time series of the different reconstructions overlap with each other well, even for reconstructions with few modes. The dynamic aspects of the effective velocities calculated with different mode numbers are similar with only small variations between the reconstructions with $M = 1$ and $M = 10$ modes. The low frequency characteristics of the $u_{\text{eff}}$ time series are reproduced with reasonable accuracy with just one single mode. The addition of more modes to the reconstruction introduces a few high frequency variations to the time series. However, this is not the case for vertical shear or yaw misalignment. With the addition of the second and third modes to the reconstruction, we observe that the dynamics of the vertical shear are captured, albeit a little over-predicted. For $M = 5$ or higher, good agreement between the direct determination and the ROM is obtained for all three parameters.

### 3.4 An interpretation of the POD modes

The different standard and dynamic errors $\epsilon_{\text{std}}$ and $\epsilon_{\text{dyn}}$ defined in Subsection 2.4 are used to quantify the reconstruction quality, as depicted in Figs. 11 (a) and (b). For our particular inflow scenario, we observe the following trends. The standard error of reconstructed rotor effective wind speed $u_{\text{eff}}$ remains less than 5% for the reconstruction based on the first mode, indicating



the first mode being highly correlated to the rotor averaged wind speed. For this particular inflow case, the standard reconstruction errors with 1 mode for $s_v$ and $\delta_h$ are quite high but drops when additional modes are chosen. The standard errors for all the three wind field parameters decrease below 10% when considering reconstruction with the first 4 modes, exhibiting the discontinuous behaviour when certain modes are considered. The steep fall in the standard error magnitude with the addition of certain modes indicate that the lower modes are strongly correlated with $s_v$ and $\delta_h$.


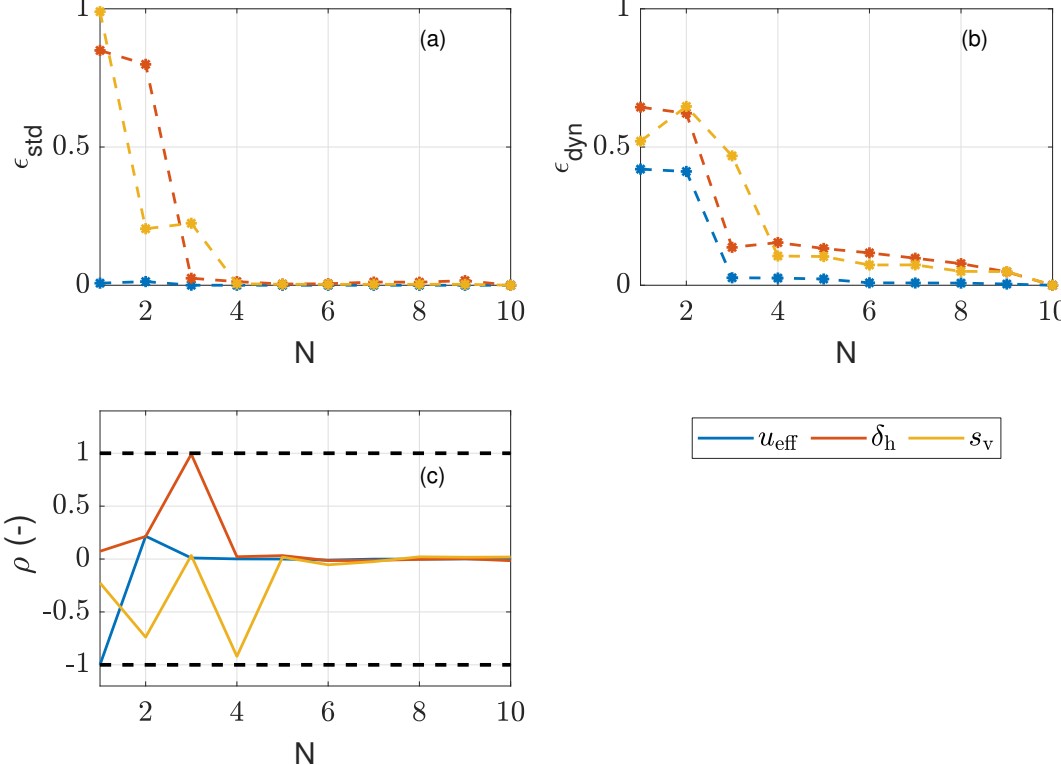

**Figure 11.** The standard ($\epsilon_{\text{std}}$) and dynamic ($\epsilon_{\text{dyn}}$) errors of the three-parameter wind field model calculated for the reduced order model with various number of modes and correlations ($\rho$) between the time series of the reconstructed three-parameter wind field parameters and the time evolution of the different POD modes $Z_j(t)$.

The dynamic error for $u_{\text{eff}}$ drops below 3% when at least the first three modes are added. The dynamic error of $s_v$ reduces to 10% with the addition of the first four modes. The dynamic error of $\delta_h$ declines below 10% with the addition of the first seven modes. Both the standard and dynamic error fall below 0.5% for all wind field parameters by including 10 modes in the reconstruction. The discontinuous behaviour shown by the standard and dynamic error occurring at identical mode numbers

indicates a relationship between the particular mode and the corresponding wind field parameter.

The huge reductions in the standard and dynamic errors can be explained by the plots of correlation between the time series of the POD time coefficients $Z_j(t)$ and the wind field parameters $X(t)$ as visualised in Fig. 11 (c). Confirming the





hypothesis based on the time series reconstruction of the rotor effective velocity (Fig. 10), the first mode is highly corre-

lated with spatial fluctuations in the wind field with $|\rho(Z_1, u_{\text{eff}})| = 0.99$. The yaw misalignment ($\delta_{\text{h}}$) is related to the third
mode with $|\rho(Z_3, \delta_{\text{h}})| = 0.99$. The vertical shear ($s_{\text{v}}$) is related to the second and the fourth modes $|\rho(Z_2, s_{\text{v}})| = 0.72$ and
$|\rho(Z_4, s_{\text{v}})| = 0.91$. The argument of the relationship between the wind field parameters and the modes is also strengthened by
the shapes of the POD modes (Fig. 8): the third and fourth mode exhibit horizontal and vertical structures while the first mode
resembles the $v_{\text{los}}$ variations.

### 3.5   Evaluation of the ROM in the frequency domain

The dynamic loading induced on the turbine blades is significantly determined by the fluctuations of the longitudinal wind
velocity impinging on the blades and how quickly they rotate. As the blades move through the turbulent wind field, they
perform a so-called 'rotational sampling' of the turbulent structures, which differ from the velocities observed at a stationary
point (Kristensen and Frandsen, 1982). To investigate this effect we calculate the autospectral density of the line-of-sight wind
speed of the reduced order model for the stationary hub centre and rotating reference frames for a radial position of 90% on

the first blade. The ROM calculated spectra are evaluated with respect to the turbulent spectra directly determined by the lidar
measurements and the reference LES (Fig. 12).

The spectra were calculated via Welch's modified periodogram method with a Hanning window and 300 s data segments
and a 50% overlap between segments. For the hub centre point visualised in Fig. 12 (a), the spectra of the different reduced
order models exhibit very similar behaviours. The lidar measurements for all cases show a remarkable drop-off from the -5/3

Kolmogorov slope at 0.03 Hz, evident of the low-pass filtering effect of the lidar. The spectrum calculated for the reduced order
model with the first mode underpredicts the energy content by one order of magnitude while the addition of more modes pushes
the spectrum upwards toward the full lidar measurements. The spectra of the reduced order model with ten modes and the full
lidar measurements show no difference, indicating that with the first ten modes, almost all energy is recovered in comparison
to the full lidar measurements. However, completely different behaviour is observed when examining the rotationally sampled

spectra of the projected longitudinal wind speed ($\omega = 11.88$ rpm) shown in Fig. 12 (b). The light blue line shows the spectrum
of the reference LES simulations sampled at 5 Hz with the 1P frequency (0.198 Hz) and the next five peaks of the higher
harmonics being clearly visible. The ROM reconstruction with $M = 1$ underpredicts and cannot capture the magnitude of the
1P and 2P peak as the first mode only reconstructs the almost rotationally symmetric wind speeds in the radial direction. With
the addition of the second and third mode, the magnitudes of the 1P and 2P peaks are accurately reconstructed while adding

more modes leads to a marginally better prediction of these rotational harmonics. The higher harmonics seen in the rotational
spectrum of the LES curve are not captured by the SpinnerLidar due to the limited 1 Hz sampling rate yielding a corresponding
Nyquist frequency of 0.5 Hz. The superior behaviour of the ROM's even $M \geq 3$ modes in the rotational spectra in comparison
to the fixed point spectra can be attributed to the high spatial coverage of the lidar. Moreover, the fact that the dominating POD
modes have clear relationships to the wind speed changes in the horizontal and vertical directions allows for better calculation

of the rotational spectrum compared to the fixed frame. Note that the SpinnerLidar measures more energy than the LES field at





lower frequencies due to cross-contamination of the lateral and vertical components contributing to increased variance in the line-of-sight velocity.

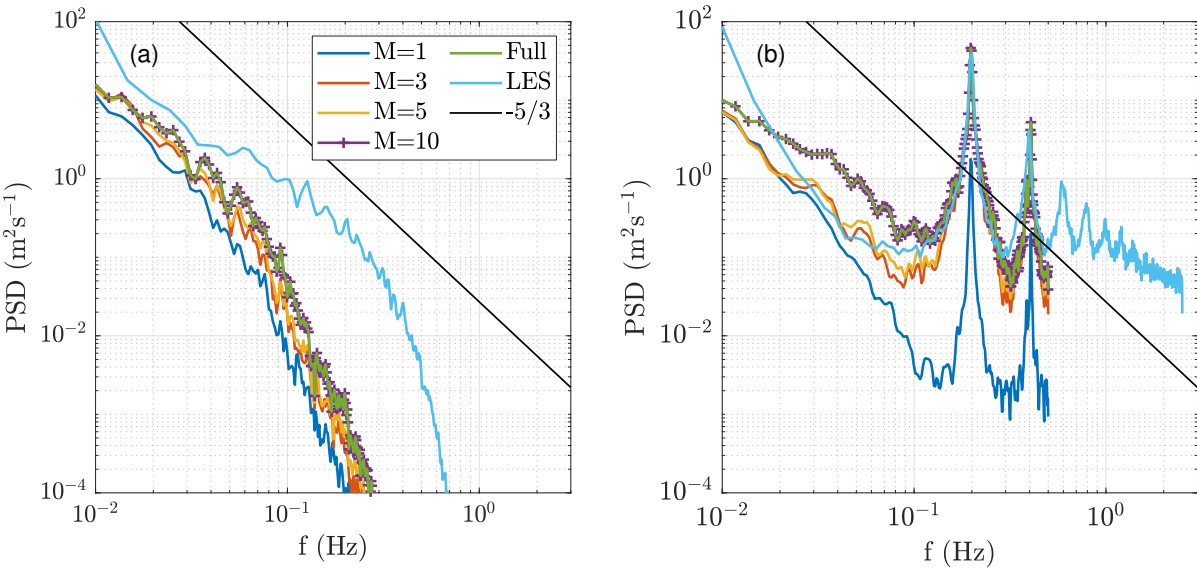

**Figure 12.** (a) Power spectral densities from the $v_{\mathrm{los}}$ velocity on the rotor axis of the reduced order model with $M = 1, 3, 5, 10$ modes at an upstream measurement distance of 126 m. LES refers to the spectrum calculated from the reference wind field sampled at 5 Hz and the Kolmogorov slope is illustrated as the black line. The spectrum directly determined from the full lidar measurements is illustrated by the green line. (b) Power spectral densities from the $u_{\mathrm{projected}}$ velocities (Eq. 1) of the reduced order model with $M = 1, 3, 5, 10$ modes at an upstream measurement distance of 126 m sampled at 90% of the outer blade radius moving at 11.88 rpm.

## 4 Discussion

A reduced order model of the wind turbine inflow was created using POD based on the inflow data from a 3700 s long interval
in a large eddy simulation. The main goal of this work was to apply POD to inflow wind fields measured by a turbine-mounted SpinnerLidar and to identify the most energetic and dominating modes which can be used to create a reduced order inflow model. By doing so, the application of low dimensional modelling methods to the temporal dynamics of the POD coefficients is possible. As our goal was to obtain a reduced order representation of the inflow, metrics related to the entire wind field were chosen. Our method provides a trade-off between conventional wind field reconstructions that require strong wind field
assumptions and time-consuming physics-based solvers. Even though this study could benefit from full 3D wind fields, POD analysis of the line-of-sight measurements from a turbine-mounted lidar capable of high spatial coverage is considered reasonable because (1) the line-of-sight measurements are dominated by the longitudinal velocity component for turbine-mounted lidars (Harris et al., 2007): (2) it is quite challenging to extract the full 3D wind field information from line-of-sight measurements (Kidambi Sekar et al., 2018) and (3) the longitudinal wind component is assumed to be the main driver of the dynamics



of turbine response dominating over the contributions of the lateral and vertical components when the turbine is aligned with the inflow.

To evaluate the performance of the reduced order model, we define the first metric as the POD mode convergence. The convergence of the eigenvalues and mode shapes (Figs. 5 and 6) indicate that a dataset with around 3500 to 4000 samples, corresponding to approximately 1 hour for the SpinnerLidar sampled at 1 Hz, should be able to provide converged results and

deterministic modes. Due to the relatively fast computation (30 s on a laptop), the analysis can be performed on the fly using a moving window to obtain POD modes which correspond to variable inflow conditions that a turbine experiences in operation. Using converged POD modes, it is feasible to develop reduced order inflow models capable of capturing the essential inflow features relevant to the turbine response.

The second metric is the cumulative energy distribution across the modes which is useful for determining the truncation point

of the reduced order model. The results indicate that the first few modes contain the most energy in the wind field. In the LES data, the first 10 modes contribute 96.6% of the total energy measured by the lidar while with 100 modes, 99.95% of the total energy is recovered (Fig. 7). The first mode contributes more than half of the total measured energy in the wind field by being highly correlated to the mean inflow wind speed. This is because the longitudinal velocity component, which dominates the line-of-sight measurement, is strongly correlated over the various spatial scales, as reported in Saranyasoontorn and Manuel

(2005) and Kidambi Sekar and Kühn (2017). The sharp slope of the eigenvalue distribution also indicates that most energy is concentrated in large scale structures. Most of the measured energy is concentrated in the first few modes as the energy associated with the small scale turbulence is effectively low-pass filtered due to the inherent volume averaging property of the lidar. This effect also nullifies the apparent disadvantage of the POD methodology whereby a large number of modes are generally required to capture small turbulent structures. Hence, the method utilizes the lidar's inherent limitation to its advantage.

Note that the lidar does not capture the total kinetic energy in the wind field as some of it is lost due to (1) directional bias on the line-of-sight wind component and (2) the volume averaging effect. Kidambi Sekar et al. (2020) estimated that for the same dataset, at an upstream focus distance of 126 m, approximately 5% of the TKE in line-of-sight velocity is lost due to the volume averaging effect.

With the truncation point at 10 modes determined initially based on the eigenvalue distribution, we define the third metric as

the data compression achieved due to the truncation itself. As the ROM eliminates modes that do not contribute significantly to the flow, only the most energetic modes are taken for reconstruction. From the LES, data compression rates of 99.5% are achieved while retaining approximately 90% of the measured TKE with a reduced order representation of 10 modes. This is a significant data reduction that is beneficial for developing real-time turbine control algorithms by compressing the inflow information into just a few signals.

The fourth metric is a qualitative one related to the reconstruction of the velocity field from the ROM. With a single mode, the reconstruction only captures the variation of the rotor effective wind speed in the measurement plane. Taking more modes into account introduces more spatial variation in the velocity field (Fig. 9). With ten modes, we obtain very good visual agreement with full lidar measurements. The model outperforms the standard three-parameter WFR of the velocity field which is unable to resolve the localised spatial structures.





The accuracy of the reduced order model is quantified by calculating wind field parameters commonly used for turbine IPC control which are also evaluated as metrics. The time series of the reconstructions with different number of modes are presented in Fig. 10 for the LES case. The low frequency characteristics of the three commonly used wind field parameters (rotor effective wind speed, vertical shear and horizontal misalignment) can be captured by the first few modes alone. The high frequency fluctuations in the wind field parameters time series were reconstructed with high accuracy by taking the first ten POD modes.

To quantify the reconstruction accuracy of these three parameters, two definitions of standard $\epsilon_{\mathrm{std}}$ and dynamic $\epsilon_{\mathrm{dyn}}$ errors are introduced. The behaviour of these errors for the simulated lidar measurements is shown in Fig. 11, which indicates their change in $\epsilon_{\mathrm{std}}$ and $\epsilon_{\mathrm{dyn}}$ as a function of number of modes. As expected, the addition of modes to the reconstruction decreases the error and causes both $\epsilon_{\mathrm{std}}$ and $\epsilon_{\mathrm{dyn}}$ to decline almost to zero when the first ten modes are taken into account. Interestingly, the addition of certain modes to the reduced order model decreases the values of the standard and dynamic errors substantially,

leading to a better representation of the wind field. For instance, the addition of the second and fourth modes reduces the standard and dynamic error associated with vertical shear quite substantially. The same behaviour is seen for the third mode, whose addition to the reconstruction reduces the estimation error in the horizontal misalignment. This suggests that certain modes play a major role in the definition of specific wind parameters while the specifics of the relationship itself will depend on the variability in the inflow conditions. The relationship between the modes and the three wind field parameters was quantified by

calculating the total correlation between the time evolution of the modes and the wind field parameter itself as seen in Fig. 11. Very clear relationships between certain wind field parameters and the time evolution of the particular modes are seen from their correlation. While these relationships will change between datasets, it is significant that only the lower modes are strongly related to the three wind field parameters. This could be exploited and the corresponding modes could be chosen with respect to a particular application by focusing on certain parameters based on either the inflow or the turbine response.

The efficiency of the POD reconstruction in the frequency domain was also investigated by calculating the turbulence spectra in the fixed and the rotating reference frames for different reduced order model reconstructions (Fig. 12). For the stationary hub height spectra, the lower order models show very similar behaviour across the whole frequency range with the SpinnerLidar measurements deviating from the Kolmogorov slope at the probe length induced drop-off frequency of 0.03 Hz. The sampling in the rotational frame of reference differs compared to the fixed frame as the large spatial coverage of the SpinnerLidar

captures the periodic fluctuations in the sampled wind speed regardless of the volume averaging effect (Kidambi Sekar et al., 2020). In the rotational spectra, the reconstruction with the first mode is incapable of predicting the magnitudes of the 1P and 2P loads in contrast to the ROM with three or more modes. As more modes are added to the reconstruction, more spatial variation in the wind field will be captured, leading to a better representation of the eddy slicing effect from which the fluctuating blade loads could be modelled.

Creating an inflow representation based on the POD methodology offers certain advantages over existing WFR methods. This method does not require strong assumptions about the wind field unlike Kapp (2017) and Raach et al. (2014) and can calculate the full wind field information instantaneously if suitable POD modes are available. This makes it attractive for wind turbine control. The location of the upstream scan distance is a very important topic of research as it is important to model the turbine's induction slowdown and wind evolution to make optimal use of turbine-mounted lidar data. An investigation by Mann et al.





(2018) concluded that the low frequency wind speed fluctuations (up to 0.006 Hz) present in the inflow are relatively unaffected by the presence of the rotor until they are very close to the rotor plane ($\leq 0.5D$). Hence, the first POD modes and their corresponding time evolution (which are related to these large scale structures) can be considered to be unaffected in the induction zone. This could be especially exploited to perform inflow measurements for wind turbines with rotor diameters larger than the maximum scanning area of the SpinnerLidar. The large scale structures related to the wind inflow parameters can be

reconstructed with high accuracy even when measurements are performed closer to the rotor. This analysis can be extended to include different atmospheric conditions (wind speeds, turbulence intensity, shear, stability, transient gusts) and operating conditions (rotor axis tilt, dynamic yaw misalignment, full and partial wake impingement).

To improve the robustness of the results, additional analysis should be carried out using simulations and data from full field experiments for a range of inflow conditions. As previously mentioned, the WFR's quality depends directly upon the lidar data

quality and is thus subject to inaccuracies caused by the device limitations itself. These shortcomings which are inherent device properties must be investigated in detail based on its potential lidar based application. In this study, the metrics for quantifying the accuracy of the model were chosen based on the inflow wind field itself. To further investigate the relationship between lidar measured wind fields and turbine dynamics, a detailed evaluation of the POD model can be performed by choosing quantities that describe the turbine's response. In further investigations, it would be beneficial to choose metrics from the turbine response

to quantify the ROM performance as suggested in Saranyasoontorn and Manuel (2005) and Bastine et al. (2018).

## 5 Conclusions

Turbine-mounted lidar measurements can be used to derive information about the inflow to the wind turbine which can subsequently be used for turbine control, loads validation or turbulence characterisation. As lidar capabilities improve due to improved hardware and larger datasets, it is crucial to reduce the measurement data to a few variables that can still capture

the spatio-temporal dynamics relevant for describing the wind field. Such models will offer a trade-off between the simple wind field reconstruction methods that require certain wind field assumptions on the one hand, and the complex CFD-based reconstructions on the other. Here, we have suggested a Reduced Order Model (ROM) for turbine-mounted SpinnerLidar measurements of the turbine inflow, based on Proper Orthogonal Decomposition (POD). The inflow model was tested with virtual SpinnerLidar measurements performed in an LES wind field. Well defined inflow modes are obtained and around 90% of the

turbulent kinetic energy measured by the SpinnerLidar in the line-of-sight direction in the wind field is captured with just the first ten POD modes. This very strong dimensional reduction indicates that the development of simplified inflow models is possible. Our method provides a way to capture most of the spatio-temporal flow information with just ten modes leading to a data reduction of 99% in comparison to the full lidar measurements for our investigated case. The velocity fields reconstructed with these dominant modes agree well with the full lidar measurements providing a method for extracting local spatial struc-

tures in the inflow. We demonstrated that certain modes are closely related to common wind field parameters such as the rotor effective wind speed, vertical shear and horizontal misalignment even though the exact relationship will differ on a case by case basis. The data reduction was possible due to the volume averaging effect of the lidar, filtering out smaller turbulent structures





deemed of lower importance for the overall turbine response, thereby taking advantage of one of the lidar's limitations. This method provides more information than classical wind field reconstruction methods: for instance, it captures the rotationally

sampled wind field and the associated first and second harmonics (1P, 2P), which dominate the dynamic blade loading quite well. The inflow model introduced here seems to be applicable to other scanning lidar systems capable of scanning the wind field with a sufficiently high spatio-temporal resolution. The method is also scalable, with respect to the evolution of more powerful lidar systems capable of even higher spatio-temporal resolution scans or better optical systems resulting in smaller probe lengths. Based on the results in this paper, our method is considered to have potential uses for lidar-based wind turbine

control, loads validation and turbulence studies.

*Data availability.*    The data of the LES simulations can be made available on request.

*Author contributions.*    APK designed the research, developed the methodology, performed the LES simulations, performed the data analysis and wrote the paper. MvD developed the analysis toolbox for the SpinnerLidar together with APK, implemented the three-parameter WFR method, and provided thorough reviews of the manuscript. AR and MK contributed with intensive discussions on the scientific content and

reviewed the manuscript thoroughly. MK supervised the research.

*Competing interests.*    The authors declare no competing interests.

*Acknowledgements.*    The research was carried out in the framework of the "DFWind" (Deutsche Forschungplattform für Windenergie) project funded by the German Federal Ministry for Economic Affairs and Energy (BMWi) based on a decision of the German Bundestag (Grant. No. 0325936C). We acknowledge the help of Sonja Krüger and Dr. Gerald Steinfeld in setting up the LES simulations and Dr. David

Bastine regarding the POD method.



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
