# Peer review of "Modelling the Wind Turbine Inflow with a Reduced Order Model based on SpinnerLidar Measurements"

_Wind Energy Science, 2021_

## Referee Comment (RC2)

**Review of the manuscript wes-2021-16, entitled "Modelling the wind turbine inflow with a reduced order model based on SpinnerLidar Measurements", by A.K. Sekar, M.F. van Dooren, A. Rott, M. Kuhn.**

This manuscript deals with the POD analysis of synthetic wind data, specifically lidar radial velocity, obtained with the virtual lidar technique applied to a single LES dataset. A truncated POD base is then used to approximate the time-series of wind parameters typically used for wind turbine control.

The authors have nicely shown how they master the use of POD on wind synthetic data; however, I have some comments on this work:

A. How this study is representative for the broad range of atmospheric and wind conditions experienced by a wind turbine? Specifically, the daily cycle of atmospheric stability leads to significant variations in velocity integral length scale, and energy distribution across scales and heights. The POD modes will vary significantly for the different conditions, and different POD modes may dominate specific conditions. Furthermore, a different number of POD modes might be needed to reconstruct a certain wind condition. Therefore, I am not sure about the applicability of this approach for real wind energy applications.

B. I am not sure about the predictive capabilities of this POD approach. In practice, the authors have carried out post-processing of wind data without providing any prediction for next-time occurrences. The authors mentioned that they plan to use this technique "on-the-fly". Even assuming that this would be computationally feasible, why you want to approximate the wind parameters with POD when you can already estimate them from the actual lidar data, and maybe with less computational costs?

I would also add that writing should be significantly improved throughout the manuscript. Several sentences should be rephrased and there is a large number of typos. Please find below some comments, which might help for further revisions.

**Comments:**

1. L 11, "...*we find that a 10 mode ROM could accurately describe most spatio-temporal variations in the inflow*". Can you comment on how this statement can be generalized for different wind/atmospheric conditions, and to rotors with different diameters and, thus, affected by structures with different sizes?

2. L 12-14, "*The reduced order modelling was accomplished using the inherent volume averaging property of lidar devices that attenuates high frequency turbulence with lower importance for the overall turbine response thus allowing significant data compression*". I have two comments on this statement: a) I am not sure how the spatial averaging of the lidar is connected with the ROM accuracy, at least not from this statement; 2) Recent works on lidar spatial averaging, see e.g. Cheynet *et al.*, Remote Sens., 2027; Puccioni & Iungo, AMT 2021, have shown that the variance of the radial velocity can be even halved from its actual value depending on lidar range gate, wind conditions and sampling height. Therefore, I would disagree that under-estimation of wind turbulence due to the lidar spatial averaging is of lower importance for the turbine response. Please comment.

3. L 18, "*have attracted greater attention*", add some representative references. Similarly at L 21 "*feed-forward lidar-assisted control*"
4. L 22, maybe fiber-based.
5. L 31, "*with high spatial and temporal resolutions*", provide some reference values and references.
6. L108, "*very high spatial and temporal*". Quantify these lidar features and provide references.
7. L123, "for small yaw misalignment and tilt angles". 30 degrees is not a small angle. I understand the simplification of neglecting *v* and *w*; however, remove the statement that this is doable based on the small lidar angles. It's only an approximation with a certain error.
8. Fig. 4, Please report figure coordinates in non-dimensional fashion with *D*.
9. L 161, you mentioned that this LES case corresponds to an unstable atmospheric regime. Then, I assume you imposed and/or quantified the respective Richardson number, Obukhov length, and surface heat flux at the terrain. Please provide these specifications of the LES.
10. Sect. 2.2, I believe that important details on the sampling of the spinner lidar from the LES dataset are missing. I believe that the spinner lidar samples much faster than for the LES sampling frequency of 5 Hz. How did you deal with the different sampling frequencies for the various lidar beams? Furthermore, how the lidar spatial averaging is implemented in the virtual lidar?
11. L 186, the square of the norm of *V* is not its TKE, rather the square absolute value. You should first define the velocity mean according to the Reynolds-averaging approach, as you are doing next with *V'*
12. L 187, Please provide details on how you define the mean velocity field.
13. L 274-275, again, this is not TKE. If you remove the mean of the flow, you will see that the energy captured by the first 10 POD modes will be much smaller than 96.6%.
14. L 360, What is the projected longitudinal wind speed, and why it is connected with the rotor speed?
15. L 390 – 394, You are suggesting using POD in real-time "on-the-fly" while collecting lidar data. So my question is, why do you want to approximate the wind parameters ($u_{eff}$, $s_v$, $delta_h$) with POD when you can estimate them directly from the lidar data, and with less computational costs? Am I missing something?

**References**

Cheynet, E., Jakobsen, J., Snæbjörnsson, J., Mann, J., Court- ney, M., Lea, G., and Svardal, B.: Measurements of surface-layer turbulence in a wide Norwegian fjord using synchronized long-range Doppler wind lidars, Remote Sens., 9, 977, https://doi.org/10.3390/rs9100977, 2017.
Puccioni, M., Iungo, G.V., Spectral correction of turbulent energy damping on wind LiDAR measurements due to spatial averaging, Atmos. Meas. Tech., 14, 1457-1474, 2021.

---

## Author Comment (AC1)

**Modelling the Wind Turbine Inflow with a Reduced Order Model based on SpinnerLidar Measurements**

**A. P. Kidambi Sekar, M. F. van Dooren, A. Rott, M.Kühn**

July 8, 2021
* * *
We thank both the reviewers for their critical assessment of our work. Before answering all the comments in detail, we summarise the changes made in the revised manuscript. The major changes are the following:

- Section 2.2 containing the description of the LES wind field has been completely rewritten to be self-contained and the simulation parameters for a second LES wind field with stable stratification has been added.

- Section 2.4 (Quantifying the Accuracy of the Reduced Order Model) has been split and rewritten in

  1. Section 3.4.1, where a detailed description of the three-parameter model is provided.
  2. Section 3.4.3, where a justification of the selection of the three-parameter model to assess model accuracy and to provide interpretation of the modes is discussed.

- Parts of Section 3 have been rewritten and several figures have been extended to also include the stable inflow case results and an inter-comparison with the unstable case results.

- Section 4 discussion has been updated with

  1. The results of the stable LES wind field.
  2. Advantages of a POD based wind field reconstruction model over parameterisation models.
  3. Applicability of POD based strategies in practice.

- An Appendix A has been added where the practical application of the model is described along with a discussion on the predictive capabilities of the method.

Smaller changes are addressed at the end of minor comments. A revised version of the manuscript is available along with a document to track changes between the old and the revised version. We hope these changes will positively benefit the manuscript.

**Reviewer 1**

In their paper "Modelling the Wind Turbine Inflow with a Reduced Order Model based on SpinnerLidar Measurements", the authors present a POD-based representation of the turbine inflow from virtual lidar measurements. The paper is original and of interest to the community. The paper is well-written and easy to read. The conclusions are properly supported by the findings. My main comment however is that it is not clear to me how the presented methodology could be used in practice, and that the presented work is focused on exploring the low rank structure from SpinnerLidar measurements rather than resulting in a practical inflow reconstruction method. Furthermore, the paper is based on a single simulation setup, which is lacking some information for reproducibility. The paper would benefit from a clarification in these areas. This is further explained in my comments below.

*We thank the reviewer for their critical assessment of our work. In the following we address their concerns point by point. Comments to the reviewer points are made in* blue.

**Main Comments**

**Comment 1**: It is not clear to me how the model could be used to reconstruct turbine inflows in practice. Reconstruction capabilities of the POD model are shown solely for a data set which the modes were fitted on, so one could say the authors do not present a model but rather the presence of low-rank structure/ compressibility of the lidar data as the result of a fitting/interpolation exercise. This is of course valuable information, but the true merit of such a ROM would lie in the application and performance for unseen data.

**Reply**: Our main objective of the paper was indeed the identification of spatial modes from which a low-rank description of the wind turbine inflow could be obtained. This model reduction would be the first step necessary to develop simplified dynamic inflow models. We try to identify modes that capture important flow aspects while yielding the necessary dimensional reduction as the accuracy of the reduced model depends on its dimensions. By performing upstream velocity measurements from the wind sensor, the temporal evolution of the POD modes can be estimated in advance if the mode shapes are known.

The ability of the POD modes to approximate complex flow is dependant on the information contained in the snapshot data required to generate the POD modes. As wind turbines operate in the atmospheric boundary layer characterised by continuously changing atmospheric and wind conditions, special considerations must be given to adopting the modes to account for inflow changes. This can be achieved through two approaches as first laid out by Bergmann et al. [2005]. The first approach would be to uniformly distribute the ensemble snapshot velocity matrix among the range of atmospheric conditions that define the inflow. This approach requires many simulations of the higher dimensional CFD model or a large SpinnerLidar dataset for generating snapshots. The second approach describes an adaptive method in which new snapshots are regularly collected and the new POD modes are calculated when the effectiveness of the existing POD ROM to represent the inflow becomes insufficient. Both these

approaches can be applied to the SpinnerLidar data that is collected in real-time operation. As the daily flow variations due to stability changes are relatively slow processes, the modes can be recalculated in longer moving time intervals. As the computational expense of calculating POD modes for a particular inflow state is in the order of a few seconds, POD based dynamic inflow models can be used to adapt and use the SpinnerLidar sensor data during the real-time phase.
We have added Appendices A1 and A2, where we describe how this method can be used in practice and investigate the reconstruction performance for unseen data. We see that the POD model can reconstruct the previewed wind field very accurately if suitable POD modes are available while the reconstruction accuracy suffers while unsuitable POD modes are used.

**Comment 2**: Could the authors clearly indicate, perhaps using a diagram with the flow of data and information, how the proposed method could be used in practice?

**Reply**:

[Figure]

Figure 1: Flowchart of the integration of the POD ROM into the turbine control system.

Figure 1 illustrates the flow diagram of the proposed method. The turbine inflow is measured with the SpinnerLidar and the velocity snapshots are used to determine the POD modes utilising the two approaches described in Comment 1. The recorded inflow data could be used to generate the ensemble snapshot matrix to determine generalised POD modes. As large amounts of data need to be processed to cover the entire operational range, it is beneficial to perform this calculation offline and use the

generalised modes for the online control. A reduced order model can be created based on the POD modes and their temporal evolution from the line-of-sight velocity measurements of the SpinnerLidar. While using the adaptive method, it is necessary to decide when the POD ROM has to be adapted to a new inflow condition. This can be either done by verifying the accuracy of the ROM, for instance by comparing the reconstructed wind fields with the full lidar data. This step can be bypassed by recalculating the POD modes at certain predefined time intervals.

As the SpinnerLidar measurements are performed upstream of the turbine, the reduced velocity fields can be realised before the wind field reaches the rotor plane. Thereby the state of the inflow in the near future is known hence facilitating a feed-forward control mechanism that determines the operational point of the turbine based on the preview wind information and regulates the turbine operation.

We have added Appendix A1 where the integration of the POD ROM into the turbine control system is discussed along with the flowchart.

**Comment 3**: Can the authors elaborate on the three-parameter function $\hat{Y}(t)$, presented in Eq. 10.

**Reply**: The operator $\hat{Y}(t)$ refers to the simplified three-parameter function defined and validated by Kapp [2017]. This method proposes a parameterisation of the inflow wind field using 3 parameters to achieve the slightest possible deviation from the actual inflow field.

A more precise explanation of the three-parameter model is now available in Section 3.4.1 of the revised manuscript.

**Comment 4**: The merit of the proposed method seems to be mainly the fact that feeding the lidar data in a truncated POD basis, rather than feeding the entire signal, into these $\hat{Y}(t)$ functions allows the lidar data to be compressed and reduces the amount of data to be processed. Does this result in significant cost savings justifying the computation of the POD modes in the first place? Some indications on where the proposed method excels over using the full lidar data are necessary.

**Reply**: The main objective of the paper is to identify the lower rank representation of the inflow. Hence, the metrics used to quantify the accuracy of the reconstructed lower rank wind field should depend on the inflow parameters that have the most effect on the turbine operation, i.e., $u_{\mathrm{eff}}, \delta_{\mathrm{h}}$ and $s_{\mathrm{v}}$. While these parameters could be directly calculated from the lidar data, we use them instead as metrics to quantify the accuracy of the reconstructed wind field. The true benefit of this method lies in the fact that the dynamics of the spatially inhomogeneous wind field can be described with a few modes making this approach attractive for higher harmonics IPC control or trailing edge flap control Unguran et al. [2019]. Furthermore, the temporal dynamics of the weighing coefficients could be evaluated to detect local events such as gusts faster than evaluating the full lidar measurements or a simple wind field reconstruction model.

In the discussion section at p. 25, l 556-570 of the revised manuscript, we elaborate on the advantages of a POD based inflow model over other wind field reconstruction approaches and full lidar data.

**Comment 5**: Could the POD modes be used outside of the dataset which they were fitted on? Or are the reconstruction capabilities of the model only available as a post-mortem processing? The authors mention in line 390 that converged POD modes could be re-obtained on the fly relatively

easily. However, reconstructing the wind field also requires the time-evolving coefficients in Eq. 7, for which the full signal $V'(x, t)$ is again necessary.

**Reply**:  In general, POD modes cannot be used outside the dataset they were fitted on. The calculated modes are valid for a particular inflow condition and need to be updated to account for the state changes. The validity of the POD modes to represent the inflow has been addressed in Comments 1 and 2 whereby methods to update the POD modes or to obtain generalised POD modes are detailed. In the revised manuscript, we have added Appendix A2 that addresses the predictive capabilities of the method and the validity of POD modes to represent flow states outside which it was calculated. In general, with the availability of appropriate modes, the inflow wind field is predicted with high accuracy while the usage of wrong POD modes to describe the inflow introduces prediction errors.

**Comment 6**:  The authors build their argumentation based on a single simulation setup of an NREL 5MW turbine with a DTU SpinnerLidar in an unstable boundary layer.
Have the authors considered adding a second lidar to mitigate the cyclops dilemma? Please comment on this in the paper.

**Reply**:

- *Single LES wind field*: We have included one more LES wind field of stable stratification to extend our argumentation and investigate the model performance for two stability cases. We also address this in Comment 7.

- *Cyclops dilemma*: We did not consider multiple lidars. In principle, it should be possible to extend the approach to multiple lidars, e.g., ground-based or integrated in the blades. While having an additional ground-based lidar or lidars scanning synchronously with the turbine-mounted SpinnerLidar could be used to resolve the 2D or even 3D velocity components, we constrain our analysis only to velocity measurements of a turbine-mounted SpinnerLidar as lidar-based feed-forward systems rely on a single turbine-mounted lidar measuring the inflow.

**Comment 7**:  The details of the precursor simulation could be improved. The paper would benefit from including information of the following:
- The authors use unstable stratification, what is the surface heat flux? Why did the authors opt for unstable stratification?
- How are the precursor simulations initialized? (temperature + velocity?)
- What is the boundary layer height after the spinup time? (Could you include snapshots of the precursor velocity field? (x,y) ; (x,z))
- p. 159, what do the 'default settings' of PALM imply? Please make the description of the setup self-contained.
- I'm assuming the precursor is periodic in x and y, please confirm.

**Reply**:  Section 2.2 in the revised manuscript has been completely rewritten to make the description, simulation setup and the characteristics of the two LES wind fields self-contained.

1. We chose an unstable stratification as the amount of turbulent kinetic energy is larger in comparison to neutral and stable flows while the most energetic motion also occurs on larger scales

[Vollmer et al., 2016]. Due to the larger energy content and convective mixing, this case would require more POD modes to capture all the TKE in the wind field. The kinematic sensible heat flux at the surface was fixed at 0.023 $Kms^{-1}$.

2. The simulation is initialised with vertically constant geostrophic winds, cyclic lateral boundary conditions, temperature profiles and surface heat fluxes. The wind speed and wind direction variation with height are obtained due to a combination of Coriolis forces, ground friction and stratification after several hours of spinup time. We specify a geostrophic wind $u_{\mathrm{g}} = 11.81$ m/s, $v_{\mathrm{g}} = $ -1.12 m/s and a surface potential temperature of 290 K.

3. The boundary layer height for the unstable case is 1416 m at the end of the spinup time. As one more LES case to the manuscript, we have decided not to show precursor velocity field, but instead, we updated Fig. 4 (a) to show the 1-D profiles of the horizontal wind speed, wind direction and potential temperature as a function of height for both the cases. We have also added Table 1 with the wind inflow characteristics (wind speeds, TI, veer and shear) that are more commonly used for defining the inflow.

4. We agree that "default settings" is not an apt description of the simulation set-up. The description is updated in the revised manuscript in Section 2.2.

5. Yes, the precursor run is periodic in the x and y directions. This information has been added to the revised manuscript in Section 2.2.

**Comment 8**: I feel the paper would benefit from adding a second case in neutral or stable stratification, where turbulence structure will be significantly different from the large convective structures in the present case. If however the authors expect this would not influence their findings, they should at least discuss this in detail.

**Reply**: To address the inflow variability and the effectiveness of the POD method in reconstructing the wind inflow, we have also added one stable stratified LES wind field in our analysis and discuss its results compared to the unstable case.
* * *
**Minor Comments**

**Comment 1** — - Figure 7, cumilative → cumulative

**Reply**: This has been fixed in the manuscript.

**Comment 2** — Figure 8, units

**Reply**: Units have been added to the figure.

**Comment 3** — - p. 10, l 245: having has the largest... → having the largest

**Reply**: This has been fixed in the manuscript.

**Comment 4** — - p. 10, l 245: *cov* → cov

**Reply**: This has been fixed in the manuscript.

**Comment 5** — - p. 11, l 262: "... start to converge around n = 3500 samples due to temporal correlations in the wind field." What do the authors mean by these temporal correlations causing convergence? Is this convergence caused by having snapshots that are separated by smaller time intervals when increasing the amount of samples, or are the n=1000 samples spaced equally as the n=3500 samples?

**Reply**: The convergence is tested by investigating the sensitivity of the results to the number of samples with the same time separation used for the estimation. The convergence is caused by increasing the number of snapshots equally spaced at 1 Hz which is the sampling rate of the SpinnerLidar. As the flow conditions remain relatively consistent while taking a large enough snapshot velocity set, the temporal variations in the inflow are smoothed and hence the eigenvalue magnitudes remain relatively constant.

**Comment 6** — - p. 13, l 277: The authors mention that "energy is distributed over different scales and its representation might require an enormous amount of POD modes." This is an interesting comment. There is indeed a wide range of scales, but the energy cascade with decreased energy at small scales is effectively what allows us to do LES. On the other hand, it is true that further dimensionality reduction is notoriously difficult, perhaps even more so than widely considered in literature. A good reference here would be Bauweraerts, Pieter, and Johan Meyers. "Study of the energy convergence of the Karhunen-Loeve decomposition applied to the large-eddy simulation of a high-Reynolds-number pressure-driven boundary layer." Physical Review Fluids 5.11 (2020): 114603.

**Reply**: Indeed resolving the larger scales and modelling the small scale turbulence is what makes LES possible. Thank you for pointing us towards this reference. We have added this reference to our revised manuscript.

**Comment 7** — - p. 14, l 297: The authors claim that the M=1 reconstruction only allows to properly reconstruct the mean. Since the mean is not considered in the POD decomposition (Eq. 4), I suspect it is added afterwards to the POD reconstructions of the POD modes. Does this imply that the first mode offers no significant reconstruction information at all?

**Reply**: Indeed the mean is not considered for the POD decomposition. As can be seen in Fig. 8, the first two modes are associated with the variations of the line-of-sight velocities in the scan due to the inclination angle of the laser beam with respect to the inflow. A better wording would be to say that the first two modes reconstruct the spatial fluctuations in the effective wind speed (Fig. 11). This has been updated in the manuscript in p18, l 375-376.

**Comment 8** — - p. 19, Figure 12 caption: $u_{\text{projected}}$ refers to equation 1. But is not defined there. Please be more precise.

**Reply**: The $u_{\text{projected}}$ velocity is obtained from Eq. (1) by assuming that the longitudinal wind speed $u$ is dominant over the cross-wind components.

$$u_{\text{projected}} = \frac{v_{\text{los}}}{\cos(\chi)\cos(\delta)} \tag{1}$$

This has been updated in the revised manuscript in the caption of Figure 13.

**Reviewer 2**

This manuscript deals with the POD analysis of synthetic wind data, specifically lidar radial velocity, obtained with the virtual lidar technique applied to a single LES dataset. A truncated POD base is then used to approximate the time-series of wind parameters typically used for wind turbine control. The authors have nicely shown how they master the use of POD on wind synthetic data; however, I have some comments on this work.

*We thank the reviewer for their critical assessment of our work. In the following, we address the concerns point by point. Comments to the reviewer points are made in blue while modifications to the manuscript are made in red.. A short overview of the changes in the revised manuscript is also available at the beginning of this document.*
* * *
**Comment 1**: How this study is representative for the broad range of atmospheric and wind conditions experienced by a wind turbine? Specifically, the daily cycle of atmospheric stability leads to significant variations in velocity integral length scale, and energy distribution across scales and heights. The POD modes will vary significantly for the different conditions, and different POD modes may dominate specific conditions. Furthermore, a different number of POD modes might be needed to reconstruct a certain wind condition. Therefore, I am not sure about the applicability of this approach for real wind energy applications

**Reply**: Wind turbines operating in the atmospheric boundary layer experience a wide range of inflow conditions. We agree that a study based on a single LES wind field does not cover the wide range of inflow conditions that a turbine experiences in operation. The modes will differ in shape and energy content based on the inflow conditions and this has been seen based on our analysis of other inflow situations and also from full field experimental data.
To address the inflow variability we have added one more stably stratified wind field as a second LES case to show the variations due to different stratification conditions. The practical implementation and application of the model to real wind energy applications has been added as an appendix. In Appendix A1, we discuss how a POD based reduced order model can be integrated into a feed-forward controller and propose two methods for recalculation of the POD modes with changes in inflow conditions. As changes in atmospheric stabilities are relatively slow processes, the modes could be recalculated at specific time intervals or when the ability of the modes to represent the inflow is diminished.

**Comment 2**: I am not sure about the predictive capabilities of this POD approach. In practice, the authors have carried out post-processing of wind data without providing any prediction for next-time occurrences. The authors mentioned that they plan to use this technique "on-the-fly". Even assuming that this would be computationally feasible, why you want to approximate the wind parameters with POD when you can already estimate them from the actual lidar data, and maybe with less computational costs?

**Reply**: We only use the wind parameters as metrics to quantify the accuracy of the wind field reconstruction. While calculating wind field parameters directly from the lidar data is less computationally

expensive than estimating the reduced velocity field, and then the wind parameters, the true benefit of this method lies in the spatial velocity field estimation in a reduced basis, which the parameterisation models cannot provide. For instance, the three-parameter model fits spatially constant parameters over the whole measurement area, which (i) does not take full advantage of the entire rotor plane scan provided by the SpinnerLidar, and (ii) consequently cannot detect in-homogeneous inflow situations such as partial wake impingement. A POD based model will reconstruct the in-homogeneous spatial velocity field based on a few signals. Hence, by a POD based reduced order model, more detailed wind field information can be retrieved compared to parameterisation models that can be subsequently used for higher harmonics control (e.g. 2P IPC) or trailing edge flap control [Ungurán et al., 2019].

We have added an Appendix section where we discuss the prediction capabilities of a POD based model which is possible due to the advance inflow wind measurements provided by the SpinnerLidar. In Appendix A2 we investigate the predictive capabilities for three different scenarios when appropriate POD modes to describe the inflow are available and when wrong POD modes are used. In general, with the availability of appropriate modes, the inflow wind field is predicted with high accuracy while the usage of wrong POD modes to describe the inflow introduces prediction errors.

**Comment 3**:    I would also add that writing should be significantly improved throughout the manuscript. Several sentences should be rephrased and there is a large number of typos. Please find below some comments, which might help for further revisions.

**Reply**:  We thank the reviewer for the comments to help with the manuscript revision.
* * *
**Minor Comments**

**Comment 1**  — L 11, ". . . we find that a 10 mode ROM could accurately describe most spatio-temporal variations in the inflow". Can you comment on how this statement can be generalized for different wind/atmospheric conditions, and to rotors with different diameters and, thus, affected by structures with different sizes?

**Reply**:  A general statement cannot be derived without analysing a large enough dataset that considers variations in wind/atmospheric conditions and rotor sizes. We do not have such a large dataset available. However, we see that a generalised ten mode ROM is sufficient to capture the inflow irrespective of the inflow conditions from the inflow LES situations in the revised manuscript and available free-field experimental data.

We try to approach the generalisation first from the perspective of the inflow. For the two LES wind fields of different stratification's, a reduced-order model with the first ten modes captures most of the spatio-temporal variations in the inflow. The statement holds also for SpinnerLidar measurements performed at a shorter focus distance in full field experiments. To back this statement, we show the eigenvalue distributions from two free-field measurement cases performed in 2016 and 2017 respectively. In this campaign, the SpinnerLidar measured 37 m upstream while installed on the spinner of a turbine with a rotor diameter of 114.9 m (lidar is measuring at approximately 0.32 $D$). More details of this measurement campaign can be found in Bromm et al. [2018]. We analyse the SpinnerLidar measured inflow on two days spaced half a year apart as tabulated in Table 1. The energy distribution against the mode number in Figure 2 exhibits the same behaviour seen in the LES cases as well. With 10 modes, 94% of the energy in the wind field is recovered for the unstable case and 89% for the stable case with

similar results for the other metrics which we do not show here for brevity.

|  | Measurement Period | $f_{\text{spilol}}$ (m) | $u_0$ m/s | $\delta_h$ (°) | TI (-) | Stratification |
|---|---|---|---|---|---|---|
| Case 1 | 09:00-10:00 on 15-11-2016 | 37 | 9.1 | 0.7 | 0.20 | Unstable |
| Case 2 | 09:00-10:00 on 24-05-2017 | 37 | 8.2 | 0.3 | 0.135 | Stable |

Table 1: Overview of the free field measurement cases.

[Figure]

Figure 2: Energy distribution for the two full field measurement cases.

The effect of larger scanning area required for larger turbines and the representation of larger atmospheric flow structures should be further analysed. For such an analysis, the parameters of the SpinnerLidar should be changed to increase the measurement range and decrease the probe volume at larger distances.

**Comment 2**: L 12-14, "The reduced order modelling was accomplished using the inherent volume averaging property of lidar devices that attenuates high frequency turbulence with lower importance for the overall turbine response thus allowing significant data compression". I have two comments on this statement: 1) I am not sure how the spatial averaging of the lidar is connected with the ROM accuracy, at least not from this statement; 2) Recent works on lidar spatial averaging, see e.g. Cheynet et al., Remote Sens., 2017; Puccioni and Iungo, AMT 2021, have shown that the variance of the radial velocity can be even halved from its actual value depending on lidar range gate, wind conditions and sampling height. Therefore, I would disagree that underestimation of wind turbulence due to the lidar spatial averaging is of lower importance for the turbine response. Please comment.

**Reply**:

1. The spatial averaging property of a lidar low-pass filters the high-frequency turbulence present in our measurements, i.e., turbulent structures smaller than the measurement volume along the laser

beam direction cannot be measured. Hence the filtered variance measured by a lidar is less than the unfiltered variance in the wind due to the averaging effect along the laser beam direction. For a cw lidar, Sjöholm et al. [2009], show that the lidar measured spectra is dominated by noise at high frequencies after the drop-off frequency. By neglecting the higher modes that contain contributions from the higher frequencies in the lidar measured spectrum, a spatial filtering is applied to the data. As our interest lies in structures larger than the measurement volume, this spatial filtering works in our favour exploiting the synergy between the method and the device. A more clear description of the ROM accuracy and the suitability of the ROM for representing inflow details relevant for the turbine is available at p. 25, l 571-581 of the revised manuscript. Moreover, we have changed the text at p. 1, l 12-14 to the following to make it less ambiguous:

The reduced-order modelling was accomplished using the inherent volume averaging property of lidar devices that attenuates high-frequency turbulence thus allowing significant data compression.

2. The underestimation of turbulence significantly impacts lidar-based load validation as the turbulence intensity is the primary driver of fatigue loads. Bossanyi et al. [2014] show that for control purposes, however, the volume averaging effect is actually beneficial for the calculation of rotor averaged quantities as it resembles a spatial averaging of the wind field by the rotor. With scanning lidar systems, sampling points distributed with the movement of the laser beam covers a larger rotor area than perfectly focused measurements. Moreover, due to wind evolution, tracking high-frequency turbulent structures smaller than the measurement volume from the upstream measurement location to the rotor plane is not possible. However, the lidar can easily measure the larger turbulent structures that remain relatively unchanged as they move towards the rotor. We have added the following text to the discussion at p. 24, l 515:
Please note that while the volume averaging induced turbulence attenuation is beneficial for control purposes due to the spatial averaging on the rotor area [Bossanyi et al., 2014] , it has a significant impact on lidar-based load validation as the turbulence is the primary driver of fatigue loads.

**Comment 3**:    L 18, "have attracted greater attention", add some representative references. Similarly at L 21 "feed-forward lidar-assisted control"

**Reply**:  Reference Schlipf et al. [2011] added to text.

**Comment 4**:    L 22, maybe fiber-based

**Reply**:  Text has been changed.
Substantial amount of research has been done on lidar-assisted wind turbine control following advances in photonics-based communications for fiber-based lidar technologies that emerged during the early 2000's

**Comment 5**:    L 31, "with high spatial and temporal resolutions", provide some reference values and references.

**Reply**:  Text has been changed and corresponding references have been added to the revised manuscript. With such advanced devices available, it is possible to measure the inflow of a large wind turbine with

high spatial and temporal resolutions with the SpinnerLidar capable of measuring 500 points in one second.

**Comment 6**: L123, "for small yaw misalignment and tilt angles". 30 degrees is not a small angle. I understand the simplification of neglecting v and w; however, remove the statement that this is doable based on the small lidar angles. It's only an approximation with a certain error.

**Reply**: The calculation of the projected longitudinal wind speed component is indeed an approximation. Text has been changed.
"the lateral velocities are assumed to be zero as the longitudinal component is dominant $(u \gg v, w)$ for nacelle or spinner-based lidar measurements for small yaw misalignment angles."

**Comment 7**: Fig. 4, Please report figure coordinates in non-dimensional fashion with D.

**Reply**: The axis of the figure has been non-dimensionalised based on the rotor diameter of the NREL 5 MW turbine.

**Comment 8**: L 161, you mentioned that this LES case corresponds to an unstable atmospheric regime. Then, I assume you imposed and/or quantified the respective Richardson number, Obukhov length, and surface heat flux at the terrain. Please provide these specifications of the LES

**Reply**: The values of the Gradient Richardson number, Obukhov length, and surface heat flux are 0.176, -452.03 m, 0.023 $Kms^{-1}$ respectively. This information has been added to Section 2.2 of the revised manuscript.

**Comment 9**: Sect. 2.2, I believe that important details on the sampling of the spinner lidar from the LES dataset are missing. I believe that the spinner lidar samples much faster than for the LES sampling frequency of 5 Hz. How did you deal with the different sampling frequencies for the various lidar beams? Furthermore, how the lidar spatial averaging is implemented in the virtual lidar?

**Reply**: We have added more details on the lidar simulator in Section 2.2 of the revised manuscript. The lidar simulator emulates the lidar measurements inside the LES wind field while taking account of the volume averaging effect and producing 312 measurement points distributed along the rosette scan pattern. For every measurement point, the lidar simulator freezes the wind field and performs linear interpolation to obtain the projection along the laser beam direction. The lidar spatial averaging is described as a Lorenzian function as described by Sjöholm et al. [2009] for continuous-wave lidars. From the lidar properties and the focus distance, the length of the measurement volume is defined. The wind field is then interpolated over this volume and the wind velocities along the line-of-sight are weighted based on the Lorenzian function.

**Comment 10**: L 186, the square of the norm of V is not its TKE, rather the square absolute value. You should first define the velocity mean according to the Reynolds-averaging approach, as you are doing next with V'

**Reply**: Indeed $||V(X, t)||^2$ is the square of the absolute velocity value while the fluctuating part is related to the TKE. We have removed this statement.

**Comment 11**:   L 187, Please provide details on how you define the mean velocity field.

**Reply**:   We remove the mean velocity field to obtain a snapshot matrix of the flow fluctuations. The mean is defined as the time-averaged velocity field over the SpinnerLidar measurement trajectory for the total simulation duration. We have modified the text to:
Here, $\langle V(X,t) \rangle$ denotes the spatial velocity field averaged over time.

**Comment 12**:   L 274-275, again, this is not TKE. If you remove the mean of the flow, you will see that the energy captured by the first 10 POD modes will be much smaller than 96.6%.

**Reply**:   Thank you for pointing this out. We do indeed recover the fluctuating kinetic energy in the wind field. We have rephrased this accordingly.
It is clear from Fig. 7, the first 10 modes contribute 96.6% of the turbulent kinetic energy (TKE) of the field $V'$ while the first 100 modes accounts for 99.95%.

**Comment 13**:   L 360, What is the projected longitudinal wind speed, and why it is connected with the rotor speed?

**Reply**:   The standard definition of the $u_{\text{projected}}$ velocity is obtained from Eq. (1) by assuming that the longitudinal wind speed $u$ is dominant over the cross-wind components. While this is an approximation as pointed out in Comment 9, it is a very common approach to write

$$u_{\text{projected}} = \frac{v_{\text{los}}}{\cos(\chi)\cos(\delta)}. \tag{2}$$

Collecting the projected longitudinal velocities by following a point on the blade as it slices through the wind results in a time series of the rotationally sampled velocity, i.e., the velocities the blade will experience as it slices through the wind.
We have added the definition of the projected longitudinal speed to the caption of Figure 13.

**Comment 14**:   L 390 – 394, You are suggesting using POD in real-time "on-the-fly" while collecting lidar data. So my question is, why do you want to approximate the wind parameters (ueff, sv, deltah) with POD when you can estimate them directly from the lidar data, and with less computational costs? Am I missing something?

**Reply**:   Please refer to our answer to Comment 2 where we state that we use these parameters to quantify the accuracy of the reduced velocity field. The true advantage of this method lies in the estimation of the spatial velocity field in a reduced sense which the three-parameter method does not provide. We have rewritten this part of the discussion in p. 25, l 556-570 of the revised manuscript, where the advantages of using a POD based inflow model over parameterisation models and full lidar data is discussed in detail. The challenges with the online calculation of POD modes and the predictive capabilities are explained in Appendices A1 and A2.

**References**

Michel Bergmann, Laurent Cordier, and Jean-Pierre Brancher.  Optimal rotary control of the cylinder wake using proper orthogonal decomposition reduced-order model. *Physics of Fluids*, 17(9):097101, 2005. doi: 10.1063/1.2033624.

E A Bossanyi, A Kumar, and O Hugues-Salas. Wind turbine control applications of turbine-mounted LIDAR. *Journal of Physics: Conference Series*, 555:012011, dec 2014. doi: 10.1088/1742-6596/555/1/012011.

Marc Bromm, Andreas Rott, Hauke Beck, Lukas Vollmer, Gerald Steinfeld, and Martin Kühn. Field investigation on the influence of yaw misalignment on the propagation of wind turbine wakes. *Wind Energy*, 21(11):1011–1028, 2018. doi: 10.1002/we.2210.

S. Kapp. *Lidar-Based Reconstruction of Wind Fields and Application for Wind Turbine Control.* PhD thesis, Carl von Ossietzky Universität Oldenburg, 2017.

David Schlipf, Stefan Kapp, Jan Anger, Oliver Bischoff, Martin Hofsäß, Andreas Rettenmeier, and Martin Kühn. Prospects of optimization of energy production by lidar assisted control of wind turbines. *Proc. EWEA Annu. event. Vienna*, 2011. doi: 10.18419/opus-3916.

Mikael Sjöholm, Torben Mikkelsen, Jakob Mann, Karen Enevoldsen, and Michael Courtney. Spatial averaging-effects on turbulence measured by a continuous-wave coherent lidar. *Meteorologische Zeitschrift*, 18(3):281–287, 06 2009. doi: 10.1127/0941-2948/2009/0379.

R. Ungurán, V. Petrović, L. Y. Pao, and M. Kühn. Uncertainty identification of blade-mounted lidar-based inflow wind speed measurements for robust feedback–feedforward control synthesis. *Wind Energy Science*, 4(4):677–692, 2019. doi: 10.5194/wes-4-677-2019.

L. Vollmer, G. Steinfeld, D. Heinemann, and M. Kühn. Estimating the wake deflection downstream of a wind turbine in different atmospheric stabilities: an les study. *Wind Energy Science*, 1(2):129–141, 2016. doi: 10.5194/wes-1-129-2016.